# Parameter-free Dynamic Regret:
# Time-varying Movement Costs, Delayed Feedback, and Memory

**Hao Qiu** [* 1]   **Andrew Jacobsen** [* 2 3]   **Emmanuel Esposito** [* 2]   **Mengxiao Zhang** [* 4]

## Abstract

In this paper, we study dynamic regret in unconstrained online convex optimization (OCO) with movement costs. Specifically, we generalize the standard setting by allowing the movement cost coefficients $\lambda_t$ to vary arbitrarily over time. Our main contribution is a novel algorithm that establishes the first comparator-adaptive dynamic regret bound for this setting, guaranteeing $\widetilde{\mathcal{O}}(\sqrt{(M^2 + MP_T)(T + \sum_t \lambda_t)})$ regret, where $P_T$ is the path length of the comparator sequence over $T$ rounds and $M$ is the maximal comparator norm. Our result recovers the optimal adaptive rates for both static and dynamic regret in OCO as the special case where $\lambda_t = 0$ for all rounds. To demonstrate the versatility of our results, we consider two applications: *OCO with delayed feedback* and *OCO with time-varying memory*. We show that both problems can be translated into time-varying movement costs, establishing a novel reduction specifically for the delayed feedback setting that is of independent interest. A crucial observation is that the first-order dependence on movement costs in our regret bound plays a key role in enabling optimal comparator-adaptive dynamic regret guarantees in both settings.

## 1. Introduction

Online Convex Optimization (OCO) (Cesa-Bianchi & Lugosi, 2006; Hazan, 2016; Orabona, 2025) provides a robust framework for sequential decision-making under uncertainty. In the standard paradigm, a learner iteratively selects a decision and incurs a convex loss, aiming to minimize regret—the difference between the learner's cumulative loss and that of the best fixed decision in hindsight. However, in many practical systems, changing one's decision is rarely cost-free: in domains such as optimal control (Goel et al., 2017; Goel & Wierman, 2019), video streaming (Joseph & de Veciana, 2012), and geographical load balancing (Lin et al., 2012), rapid fluctuations in strategy incur significant operational overhead, commonly referred to as *switching* or *movement costs*. Motivated by these challenges, we study a variation of online optimization in which the learner incurs an additional penalty proportional to the distance between consecutive decisions. Beyond its direct relevance to the aforementioned applications, the analysis of OCO with movement costs is instrumental in solving broader fundamental problems. A prime example is online learning with memory (Merhav et al., 2002; Anava et al., 2015), a framework designed to capture temporal dependencies in learning tasks. Reductions involving OCO with movement costs have proven essential for deriving state-of-the-art results in this domain (e.g., Anava et al. (2015); Agarwal et al. (2019); Foster & Simchowitz (2020); Zhao et al. (2023)).

On the other hand, modern online learning challenges are frequently characterized by unconstrained decision spaces and non-stationarity. A typical example is portfolio management with leverage and short-selling (McMahan & Streeter, 2012), where an agent allocates unbounded capital across assets and must adapt to shifting market regimes. In such environments, static benchmarks prove overly conservative, requiring the use of *dynamic regret* to track a moving target. Crucially, portfolio rebalancing incurs costs such as transaction fees and market impact; these costs are not constant, but rather fluctuate with market liquidity and volatility. These factors motivate a unified framework that evaluates performance against an evolving benchmark while explicitly accounting for time-varying movement penalties.

Despite this motivation, existing literature addresses these challenges only in isolation. While Zhang et al. (2022a;b) study unconstrained OCO with movement costs and Zhang et al. (2021) investigates dynamic regret with movement costs, the intersection of these two settings remains unexplored. Furthermore, all prior works assume fixed movement penalties, a restriction that fails to capture the temporal fluctuations described above. This raises the question:

---

[*]Equal contribution   [1]National University of Singapore   [2]Università degli Studi di Milano   [3]Politecnico di Milano   [4]University of Iowa. Correspondence to: Andrew Jacobsen <contact@andrew-jacobsen.com>.

*Proceedings of the $43^{rd}$ International Conference on Machine Learning*, Seoul, South Korea. PMLR 306, 2026. Copyright 2026 by the author(s).

*Can we design efficient algorithms that achieve near-optimal dynamic regret in unconstrained domains with time-varying movement costs?*

**Contributions.** In this paper, we answer this question affirmatively by developing the first parameter-free algorithm for unconstrained OCO that simultaneously handles dynamic regret and time-varying movement costs. Our primary result establishes a regret bound of $\widetilde{\mathcal{O}}\left(\sqrt{(M + P_T) \sum_{t=1}^{T}(\|g_t\|^2 + \lambda_t\|g_t\|)\|u_t\|}\right)$, where $T$ is the horizon, $g_t$ is the realized gradient at round $t$, $M$ is the maximum norm of the comparator sequence, $\lambda_t$ and $u_t$ are the movement cost coefficient and the comparator at round $t$, and $P_T$ represents the comparator sequence path length. This guarantee exhibits three key layers of adaptivity: it scales with the comparator complexity, exploits favorable geometry through realized first-order feedback, and adjusts to arbitrary fluctuations in movement penalties.

We further demonstrate the versatility of our framework via reductions to two applications, showing that OCO with movement costs can serve as a primitive for other tasks. First, we show that OCO with delayed feedback reduces to time-varying movement costs by treating the number of missing gradients as the movement scale. This reduction yields a parameter-free dynamic regret bound of $\widetilde{\mathcal{O}}\left(\sqrt{(M^2 + MP_T)(T + d_{\mathrm{tot}})}\right)$, where $d_{\mathrm{tot}}$ denotes the total delay. Compared with Wan et al. (2024), who establish a bound of $\widetilde{\mathcal{O}}\left(\sqrt{(1 + P_T)(T + d_{\max}T)}\right)$ for *bounded* domains (where $d_{\max}$ denotes the maximum delay) and only improve this to a $d_{\mathrm{tot}}$ dependence under the restrictive assumption of in-order feedback, our result handles *unbounded* domains and achieves the tighter $d_{\mathrm{tot}}$ dependence without assuming in-order arrival.

Second, under the assumption of coordinate-wise Lipschitz continuity, OCO with time-varying memory can be similarly reduced to our setting. This yields a regret bound of $\widetilde{\mathcal{O}}\left(\sqrt{(M^2 + MP_T)(H^2T + GH\sum_t b_t^2)}\right)$, where $G$ is the coordinate-wise Lipschitz constant, $H$ bounds the gradient norm of the unary losses, and $b_t$ denotes the time-varying memory length. Compared with Zhao et al. (2023), who study dynamic regret for OCO with fixed memory length $B \geq 1$ over bounded domains and establish a parameter-free bound of $\widetilde{\mathcal{O}}\left(\sqrt{(1 + P_T)(\sqrt{G}H^2B + GHB^2)T}\right)$, our result applies to unconstrained domains and improves the dependence on the time-varying memory length.

### 1.1. Related works

**OCO with movement costs.** Due to its wide range of applications, OCO with movement costs (also known as *smoothed OCO*) has been extensively studied recently (Chen et al., 2018; Goel et al., 2019; Li et al., 2020), and movement-based penalties have been studied in a variety of classic variations of OCO, including prediction with expert advice (Cesa-Bianchi et al., 2013), multi-armed bandits (Dekel et al., 2014), and bandits with feedback graphs (Rangi & Franceschetti, 2019; Arora et al., 2019). The works most directly related to ours are Zhang et al. (2022b), which establishes the first guarantees for *unconstrained static regret* with movement costs, and Zhang et al. (2021), which develops the first optimal *dynamic regret* guarantees for *bounded* domains. Our work addresses the intersection of these two challenging settings by developing the first *dynamic* regret guarantees in *unconstrained* environments with movement costs. Another closely related line of work concerns OCO with memory (Merhav et al., 2002), where the loss at each round depends on a history of past decisions. Many algorithms in this setting enforce stability via reductions to OCO with movement costs (Anava et al., 2015). More recently, Zhao et al. (2023) established dynamic regret guarantees for OCO with memory. While prior work focuses on constant memory length, our work provides the first dynamic-regret guarantees in unconstrained domains with time-varying memory.

**Parameter-free online learning.** Our work builds upon the recent line of work on comparator-adaptive (also known as *parameter-free*) methods in OCO. The unconstrained case was first studied by (McMahan & Streeter, 2012), in which a $\Omega(\|u\|\sqrt{T\log(\|u\|\sqrt{T}/\epsilon + 1)})$ lower bound was established for the unconstrained OLO setting. Later, McMahan & Orabona (2014) obtained the optimal regret $R_T(u) = \widetilde{\mathcal{O}}\left(\|u\|\sqrt{T\log(\|u\|\sqrt{T}/\epsilon + 1)}\right)$ uniformly over $u \in \mathcal{W}$. The key property of these comparator-adaptive regret bounds is that instead of scaling with the diameter of the decision set $D = \sup_{x,y \in \mathcal{W}}\|x - y\|$, the parameter-free guarantees are *adaptive* to an arbitrary comparator norm. This property is crucial in unconstrained domains where $D$ is not finite in general. These results have since been extended in various ways to achieve improved adaptivity to other problem-dependent quantities, such as achieving optimal adaptivity to the sequence of gradient norms and removing the requirement for prior knowledge of the Lipschitz constant (Orabona & Pál, 2016; Cutkosky & Orabona, 2018; Cutkosky, 2019; Mhammedi & Koolen, 2020; Jacobsen & Cutkosky, 2022; Cutkosky & Mhammedi, 2024). Our work extends this line of work by achieving comparator-adaptive bounds for three extensions of the vanilla OCO setting.

**Dynamic regret.** The extension of regret to account for a *sequence* of comparators $(u_t)_{t=1}^{T}$ was originally introduced by Herbster & Warmuth (1998a;b). The scope was later extended to the more general OCO setting in the seminal work of Zinkevich (2003), where it was also shown that Online Gradient Descent guarantees dynamic regret of $\widetilde{\mathcal{O}}(P_T\sqrt{T})$,

where $P_T = \sum_t \|u_t - u_{t-1}\|$ is the path-length of the comparator sequence. Yang et al. (2016) later showed that the bound can be improved to $\widetilde{\mathcal{O}}(\sqrt{P_T T})$ when given prior knowledge of $P_T$, and the first to achieve this guarantee without prior knowledge of $P_T$ was Zhang et al. (2018). Each of these works assumes a domain of bounded diameter. These results were later extended to unconstrained settings in various works, with a worst case bound of the form $\widetilde{\mathcal{O}}(\sqrt{(M^2 + MP_T)T})$, where $M = \max_t \|u_t\|$ (Jacobsen & Cutkosky, 2022; Luo et al., 2022; Jacobsen & Cutkosky, 2023; Jacobsen & Orabona, 2024; Jacobsen et al., 2025). Of particular note is the work of Jacobsen & Cutkosky (2022), which adapts simultaneously to both the path-length $P_T$ and the individual comparator norms, to ensure a bound of $\widetilde{\mathcal{O}}(\sqrt{(M + P_T)\sum_t \|g_t\|^2 \|u_t\|})$, achieving adaptivity to all problem-dependent quantities of interest. Our work extends this result by achieving bounds of an analogous form in the presence of an arbitrary sequence of movement costs.

**OCO with delayed feedback.** Weinberger & Ordentlich (2002); Joulani et al. (2013) first adopted black-box reductions to standard OCO to handle online learning with delayed feedback, establishing regret bounds of $\mathcal{O}(\sqrt{T + d_{\max}T})$, where $d_{\max}$ denotes the maximum delay. For convex losses, Quanrud & Khashabi (2015) refined this by adapting online gradient descent to achieve $\mathcal{O}(\sqrt{T + d_{\text{tot}}})$, where $d_{\text{tot}}$ represents the total accumulated delay, a significant improvement since $d_{\text{tot}} \le d_{\max}T$. More recently, Wan et al. (2024) extended this study to non-stationary environments, proving a dynamic regret bound of $\mathcal{O}(\sqrt{(1 + P_T)(T + d_{\max}T)})$, where $P_T$ is the comparator path length. While they improve this rate to $\mathcal{O}(\sqrt{(1 + P_T)(T + d_{\text{tot}})})$ under the assumption of in-order feedback, our work achieves this tighter dependence on $d_{\text{tot}}$ without requiring in-order arrival. Finally, there is a parallel line of work focused on obtaining adaptive regret guarantees or fast rates in delayed settings (McMahan & Streeter, 2014; Joulani et al., 2016; Wan et al., 2022; Wu et al., 2024; Qiu et al., 2025). Delayed feedback has also been thoroughly studied in the multi-armed bandit framework (Cesa-Bianchi et al., 2016; Zimmert & Seldin, 2020; Masoudian et al., 2022; Van der Hoeven & Cesa-Bianchi, 2022; Esposito et al., 2023; Van der Hoeven et al., 2023; Masoudian et al., 2024; Zhang et al., 2025; Ryabchenko et al., 2025).

## 2. Problem Setting

A major focus of this paper is the problem of online convex optimization with *movement costs*. The interaction between the learner and the environment lasts for $T \in \mathbb{Z}_+$ rounds. At each round $t \in [T]$, the learner selects $w_t \in \mathcal{W}$, where $\mathcal{W}$ is a convex decision space, while the environment simultaneously selects a convex loss function $f_t : \mathcal{W} \to \mathbb{R}$.

The learner then incurs the loss of their decision $f_t(w_t)$, with an additional penalty given by the movement cost $\lambda_t \|w_t - w_{t-1}\|$ on rounds $t > 1$. Finally, the learner observes the gradient $g_t \triangleq \nabla f_t(w_t)$ as well as the movement-cost coefficient $\lambda_{t+1}$ for the next round. The goal of the learner is to minimize the dynamic regret:

$$
\mathcal{R}_T(u_{1:T}, \lambda_{1:T}) \triangleq \sum_{t=1}^{T} \big(f_t(w_t) + \lambda_t \|w_t - w_{t-1}\|\big)
$$
$$
- \sum_{t=1}^{T} \big(f_t(u_t) + \lambda_t \|u_t - u_{t-1}\|\big) ,
$$

where $(u_1, \ldots, u_T) \in \mathcal{W}^T$ is an arbitrary sequence of comparators, and for notational convenience we extend the movement sequence by defining $u_0 = u_1$, $w_0 = w_1$, and letting $\lambda_t = 0$ for $t \notin \{2, \ldots, T\}$.

While the above definition of dynamic regret $\mathcal{R}_t$ is "symmetric" in that both the learner's and the comparator's sequence suffer from movement costs, some prior works (Zhang et al., 2021; Zhao et al., 2023) on OCO with movement costs consider the following harder objective:

$$
R_T(u_{1:T}, \lambda_{1:T}) \triangleq \sum_{t=1}^{T} \big(f_t(w_t) - f_t(u_t) + \lambda_t \|w_t - w_{t-1}\|\big),
$$

where only the learner incurs the additional penalty given by movements. Notice in particular that $\mathcal{R}_T(u_{1:T}, \lambda_{1:T}) = R_T(u_{1:t}, \lambda_{1:T}) - \sum_{t=1}^{T} \lambda_t \|u_t - u_{t-1}\|$, so the symmetric regret $\mathcal{R}_T$ is strictly easier to control. Hence, throughout this work we focus on directly bounding the harder asymmetric notion of regret $R_T$, which immediately implies bounds for $\mathcal{R}_T$ as well.

**Goal.** Our main goal is to design *parameter-free* algorithms for the above problem that guarantee dynamic regret bounds simultaneously adapting to (i) the realized sequence of observed gradients $(g_t)_{t \ge 1}$, (ii) the (time-varying) movement-cost coefficients $(\lambda_t)_{t \ge 1}$, and (iii) the complexity of the comparator sequence $(u_t)_{t \ge 1}$, measured via its *path-length* $P_T = \sum_{t=2}^{T} \|u_t - u_{t-1}\|$ and *effective diameter* $M = \max_t \|u_t\|$. Importantly, throughout this paper we focus in particular on the *unconstrained* setting, with $\mathcal{W} = \mathbb{R}^n$, which makes it substantially more challenging to obtain parameter-free guarantees that remain meaningful without *a priori* bounds on comparator norms or path length. We show that guarantees for OCO with time-varying movement costs yield clean reductions to other notable online learning problems: they imply parameter-free dynamic regret bounds for *OCO with time-varying memory* (via standard Lipschitzness assumptions) and, perhaps more surprisingly, for *OCO with delayed feedback*.

---

**Algorithm 1** Composite Mirror Descent for Movement Costs

---

**Input:** learning rate $\eta > 0$, value $\epsilon_0 > 0$
**Initialize:** $w_1 = \mathbf{0}$, $\gamma = 1/(\eta T)$, $\alpha = \epsilon_0/T$
**Define:** $\psi(w) = \frac{2}{\eta} \int_0^{\|w\|} \log\left(x/\alpha + 1\right) \mathrm{d}x$
**for** $t = 1, 2 \ldots, T$ **do**

>   Play $w_t$, then observe $g_t$ and $\lambda_{t+1}$
>   Define $\beta_t \triangleq \|g_t\| + \lambda_{t+1}$ and $\varphi_t(w) \triangleq (\eta \beta_t^2 + \gamma)\|w\|$
>   Update
>
>   $$w_{t+1} = \arg\min_{w \in \mathcal{W}} \langle g_t, w \rangle + D_\psi(w \mid w_t) + \varphi_t(w) \quad (1)$$

---

**Algorithm 2** Dynamic Parameter-free Subroutine

---

**Input:** $L > 0, \epsilon > 0$
**Initialize:** grid of learning rate choices $\mathcal{S} \triangleq \left\{ \eta_i = \frac{2^i}{L\sqrt{T}} \wedge \frac{1}{L} : i = 0, 1, \ldots \right\}$; for each $i \in [|\mathcal{S}|]$, create an instance $\mathcal{A}^{\eta_i}$ of Algorithm 1 with $\epsilon_0 = \epsilon$ and $\eta = \eta_i$.
**for** $t = 1, 2, \ldots, T$ **do**

>   Receive $w_t^{\eta_i}$ from $\mathcal{A}^{\eta_i}$ for all $i \in [|\mathcal{S}|]$
>   Play $w_t = \sum_{\eta_i \in \mathcal{S}} w_t^{\eta_i}$, then receive $g_t$ and $\lambda_{t+1}$
>   **for** each $\eta_i \in \mathcal{S}$ **do**
>   >   Send $(g_t, \lambda_{t+1})$ to $\mathcal{A}^{\eta_i}$

---

**Other notation.** Throughout the paper, we denote $\lambda_{\max} = \max_t \lambda_t$. We use $\mathbb{Z}_+$ to denote the set of positive integers and $[m]$ to denote the set $\{1, 2, \ldots, m\}$ for $m \in \mathbb{Z}_+$. Let $\mathbf{0}$ denote the all-zero vector of appropriate dimension. We use $\|\cdot\|$ to denote the Euclidean norm. For a finite set $\mathcal{S}$, we use $|\mathcal{S}|$ to denote the cardinality of $\mathcal{S}$. The Bregman divergence w.r.t. a differentiable function $\psi : \mathcal{W} \to \mathbb{R}$ is $D_\psi(x \mid y) \triangleq \psi(x) - \psi(y) - \langle \nabla\psi(y), x - y \rangle$. Given $G \geq 0$, we say that a function $f : \mathcal{W} \to \mathbb{R}$ is $G$-Lipschitz if $|f(x) - f(y)| \leq G\|x - y\|$ for all $x, y \in \mathcal{W}$.

## 3. Parameter-free OCO with Movement Costs

In this section, we propose our first parameter-free algorithm for unconstrained OCO with movement costs, summarized in Algorithm 1. Our approach adopts the Composite Mirror Descent framework (Duchi et al., 2010; Jacobsen & Cutkosky, 2022) but employs a carefully chosen regularizer. At each round $t$, after playing $w_t$ and observing the gradient $g_t$ and the coefficient $\lambda_{t+1}$, we update $w_{t+1}$ via Equation (1), which combines the Bregman divergence of a log-linear regularizer $\psi(w)$ and a correction term $\varphi_t(w)$.

The intuition behind the update in Equation (1) is to stabilize the mirror descent step. Specifically, while the regularizer $\psi$ enables the aggressive growth necessary for unbounded domains, its lack of strong convexity can lead to instability. The correction term helps to address this instability, acting as a leash pulling iterates back towards the origin in a controlled manner, which is crucial for learning stability in unbounded domains, as shown by Jacobsen & Cutkosky (2022). In our setting, we scale this penalty by $\beta_t^2 = (\|g_t\| + \lambda_{t+1})^2$ to explicitly account for the time-varying movement costs. This scaling acts as a dynamic friction coefficient: when the next movement is expensive (large $\lambda_{t+1}$) or the gradient is steep (large $\|g_t\|$), the algorithm becomes conservative and restricts updates. Conversely, when costs are low, the penalty relaxes, allowing the learner the flexibility to rapidly track changes in the environment.

Following an analysis similar to Jacobsen & Cutkosky

(2022), but with additional care to account for the movement costs, we can show that this algorithm is sufficient to provide a regret bound for our problem. In particular, we can obtain a bound on $R_T(u_{1:T}, \lambda_{1:T})$ of the form

$$\widetilde{\mathcal{O}}\left( \frac{M + P_T}{\eta} + \eta \sum_{t=1}^T \left(\|g_t\|^2 + \lambda_{t+1}^2\right)\|u_t\| \right),$$

where we recall that $M = \max_t \|u_t\|$ and $P_T = \sum_{t=2}^T \|u_t - u_{t-1}\|$, which leads to the optimal $\widetilde{\mathcal{O}}\big(\sqrt{(M + P_T) \sum_t (\|g_t\|^2 + \lambda_{t+1}^2)\|u_t\|}\big)$ regret under optimal tuning of $\eta$. However, not only would this require knowing the sequence of gradient norms and movement cost coefficients beforehand, but would also require prior knowledge of $M$ and $P_T$. This is made even harder since we consider mainly the unconstrained setting, where these two comparator-dependent quantities have no meaningful uniform upper bound.

To address these shortcomings, we employ a standard meta-algorithm technique for parameter-free OCO (Jacobsen & Cutkosky, 2022), detailed in Algorithm 2.

Specifically, Algorithm 2 maintains a group of parallel instances of Algorithm 1, each running with a distinct learning rate $\eta_i$ from a grid of candidate learning rates $\mathcal{S}$. At each round, the algorithm plays the sum of the decisions proposed by these base instances. This aggregation strategy allows us to decompose the total regret as

$$R_T(u_{1:T}, \lambda_{1:T}) \leq R_T^{(i)}(u_{1:T}, \lambda_{1:T}) + \sum_{j \neq i} R_T^{(j)}(\mathbf{0}, \lambda_{1:T}),$$

where $R_T^{(i)}$ (defined in Eq. (10)) denotes the linearized regret of Algorithm 1 instantiated with learning rate $\eta_i$. Since this bound holds for any arbitrary $\eta_i \in \mathcal{S}$, we are free to choose the $\eta_i$ which best approximates the optimal tuning, ensuring its regret scales as desired. At the same time, the regret contributions from any other instances $j \neq i$, evaluated against the all-zero comparator, each amount to an additive constant factor in light of the bound above. Given that $|\mathcal{S}| = \mathcal{O}(\log T)$, this second term only contributes an additive $\mathcal{O}(\log T)$ regret in total.

The intuitions discussed above are formalized in Theorem 3.1 below (with a complete proof in Appendix A). We remark that $L > 0$ here plays the role of the "effective" Lipschitz constant which, because of the presence of movement costs, will essentially be proportional to $G + \lambda_{\max}$.

**Theorem 3.1.** *Assume that $f_1, \ldots, f_T$ are $G$-Lipschitz convex functions. For any comparator sequence $(u_1, \ldots, u_T) \in \mathcal{W}^T$, Algorithm 2 with any $L \geq G + \lambda_{\max}$ and any $\epsilon > 0$ guarantees that*

$$R_T(u_{1:T}, \lambda_{1:T}) = \mathcal{O}\Big( \big( \epsilon \log T + \widetilde{M}_T(\epsilon) + \widetilde{P}_T(\epsilon) \big) L$$

$$+ \sqrt{ \big( \widetilde{M}_T(\epsilon) + \widetilde{P}_T(\epsilon) \big) \sum_{t=1}^{T} \big( \|g_t\|^2 + \lambda_{t+1}^2 \big) \|u_t\| } \Big),$$

*where $\widetilde{M}_T(\epsilon) \triangleq M\big(1 + \log\big(\frac{MT}{\epsilon} + 1\big)\big)$, $M = \max_t \|u_t\|$, and $\widetilde{P}_T(\epsilon) \triangleq P_T\big(1 + \log\big(\frac{4MT^2}{\epsilon} + 1\big)\big)$.*

The full proof of Theorem 3.1 is deferred to Appendix A. Observe that Theorem 3.1 recovers the optimal second-order bound for standard OCO in the absence of movement costs (*i.e.*, when $\lambda_t = 0$ for all $t$). Moreover, the bound has the desirable property of being *adaptive* to all problem-dependent quantities of interest. In particular, notice that the dominant term in Theorem 3.1 depends on the individual movement coefficients $\lambda_t$ rather than the worst-case quantity $\lambda_{\max}$. This refinement is particularly consequential for our application to OCO with delayed feedback (see Section 5), where effective movement costs fluctuate significantly with delay lengths, making a $\lambda_{\max}$-dependent bound suboptimal.

Despite the strengths of the bound, a notable limitation of Theorem 3.1 is the squared dependency on the movement coefficients $\lambda_{t+1}^2$ within the square root. This structure implies that movement costs can induce significant regret even in benign environments where the gradients $\|g_t\|$ are negligible. In the next section, we refine the algorithm to improve this dependence, a step that is also essential for achieving optimal guarantees in our later applications.

## 4. Improved Adaptivity to Movement Costs

As noted previously, while Algorithm 2 successfully adapts to gradient norms and comparator properties, its squared dependence on movement costs is suboptimal. To refine this adaptivity, we observe that in the presence of movement costs, not every gradient observation justifies an *immediate update*. Intuitively, when the movement penalty is high, the learner should only move if the first-order information has sufficient magnitude to compensate for the cost. Otherwise, it is beneficial to temporarily freeze the iterate and aggregate information until an update becomes worthwhile.

Building on this intuition, we introduce an adaptive batching meta-algorithm that yields refined dynamic regret guar-

---

**Algorithm 3** Adaptive First-order Reduction

**Input:** $L > 0, \epsilon > 0$
**Initialize:** An instance $\widetilde{\mathcal{A}}$ of Algorithm 2 with input $L$ and $\epsilon$. Denote $\widetilde{w}_1$ to be $\widetilde{\mathcal{A}}$'s first decision, $\tau = 1$, $H_1 = \mathbf{0}$.
**for** $t = 1, 2, \ldots$ **do**
    Play $w_t = \widetilde{w}_\tau$, then receive $g_t$ and $\lambda_{t+1}$
    Set $H_\tau \leftarrow H_\tau + g_t$
    **if** $\|H_\tau\| > \lambda_{t+1}$ **then**
        Set $\widetilde{g}_\tau = H_\tau$ and $\widetilde{\lambda}_{\tau+1} = \lambda_{t+1}$
        Send $(\widetilde{g}_\tau, \widetilde{\lambda}_{\tau+1})$ to $\widetilde{\mathcal{A}}$, and receive $\widetilde{w}_{\tau+1}$
        Set $H_{\tau+1} = \mathbf{0}$ and update $\tau \leftarrow \tau + 1$

---

antees, improving the movement cost dependence from $\sum_{t=1}^{T} \lambda_t^2$ to $\sum_{t=1}^{T} \lambda_t \|g_t\|$. The pseudo-code is presented in Algorithm 3. Our approach is inspired by Zhang et al. (2022b, Algorithm 3), but extends their procedure to accommodate both time-varying movement coefficients and dynamic regret, going beyond the static regret with fixed movement cost coefficient considered in their work. Both extensions require a non-trivial analysis of the original algorithm.

Concretely, Algorithm 3 *adaptively* partitions the rounds into a sequence of epochs $I_1, \ldots, I_N$. Within each epoch $I_\tau$, the algorithm plays a fixed action $\widetilde{w}_\tau$ while accumulating gradients into a buffer $H_\tau = \sum_{t \in I_\tau} g_t$. An update is triggered only when the cumulative gradient norm is sufficiently large relative to the current movement cost, specifically when $\|H_\tau\| > \lambda_{t+1}$. Once this condition is met, the algorithm passes the aggregated gradient $\widetilde{g}_\tau = H_\tau$ and the current movement coefficient $\widetilde{\lambda}_{\tau+1} = \lambda_{t+1}$ to the base learner (Algorithm 2) to generate the decision for the next epoch. Theorem 4.1 formally states the regret bound of Algorithm 3, whose proof is deferred to Appendix B.

**Theorem 4.1.** *Assume that $f_1, \ldots, f_T$ are $G$-Lipschitz convex functions. For any comparator sequence $(u_1, \ldots, u_T) \in \mathcal{W}^T$, Algorithm 3 with any $L \geq G + 2\lambda_{\max}$ and any $\epsilon > 0$ guarantees that*

$$R_T(u_{1:T}, \lambda_{1:T}) = \mathcal{O}\Big( \big( \epsilon \log T + \widetilde{M}_T(\epsilon) + \widetilde{P}_T(\epsilon) \big) L$$

$$+ \sqrt{ \big( \widetilde{M}_T(\epsilon) + \widetilde{P}_T(\epsilon) \big) \sum_{t=1}^{T} \big( \|g_t\|^2 + \lambda_t \|g_t\| \big) \|u_t\| } \Big),$$

*where $M$, $\widetilde{M}_T(\epsilon)$ and $\widetilde{P}_T(\epsilon)$ are defined as in Theorem 3.1.*

Theorem 4.1 substantially sharpens the guarantee of Theorem 3.1. Specifically, while full second-order adaptivity is known to be incompatible with movement costs (see Gofer, 2014; Zhang et al., 2022b), our bound achieves an ideal form of first-order adaptivity. The leading data-dependent term now scales with $\sum_t \big( \|g_t\|^2 + \lambda_t \|g_t\| \big) \|u_t\|$; this effectively

replaces the quadratic movement cost dependence $\lambda_{t+1}^2$ of Theorem 3.1 with the linear interaction term $\lambda_t \|g_t\|$, while preserving granular adaptivity to individual $\lambda_t$ (rather than $\lambda_{\max}$). As we show in Section 5, *this refinement is essential* for obtaining optimal regret bounds in applications such as OCO with delayed feedback. Crucially, this bound vanishes as gradients approach zero even if movement costs remain strictly positive. Furthermore, when reducing to the one-dimensional static regret setting with fixed movement costs, our result improves upon Zhang et al. (2022b) by replacing $G^2 T + \lambda G T$ with the adaptive sum $\sum_t \|g_t\|^2 + \lambda \sum_t \|g_t\|$. Moreover, previous dynamic regret results (Zhang et al., 2021; Zhao et al., 2023) need to rely on bounded domains and have similar shortcomings regarding adaptivity, leading to coarser dependencies on problem-dependent quantities. Lastly, we remark that, to the best of our knowledge, Theorems 3.1 and 4.1 are the *first* parameter-free results achieving near-optimal dynamic regret in unconstrained OCO with movement costs.

*Remark* 4.2. Algorithm 3 needs $L \geq G + 2\lambda_{\max}$, thus requiring prior knowledge of both $G$ and $\lambda_{\max}$. While knowing $G$ is fairly standard and reasonable, $\lambda_{\max}$ ultimately depends on quantities $(\lambda_t)_{t \geq 1}$ that are only sequentially observed. Fortunately, since the learner knows $\lambda_t$ before playing $w_t$, we may avoid this unrealistic prior information via a standard doubling trick on $\|g_{t-1}\| + \lambda_t$ with initial guess $L = G$.

*Remark* 4.3. We point out that the first-order movement-cost dependence in Theorem 4.1 is in fact never worse (up to constant factors) than the corresponding second-order dependence in Theorem 3.1. To see why, observe that by the AM-GM inequality we have $\sum_t \left( \|g_t\|^2 + \lambda_t \|g_t\| \right) \|u_t\| = \mathcal{O}\left( \sum_t \left( \|g_t\|^2 + \lambda_t^2 \right) \|u_t\| \right)$. A careful reader may also notice that the bound in Theorem 3.1 is stated with $\lambda_{t+1}^2$ instead of $\lambda_t^2$, though it is easily seen that this index shift can be removed by paying only an additional lower-order $\mathcal{O}(\lambda_{\max}(M + P_T))$ penalty, which already appears in the regret bound. Consequently, the first-order movement-cost bound can always be upper bounded in terms of the second-order guarantee, yet it can be significantly sharper when $\|g_t\|$ is small compared with $\lambda_t$. A complete proof of this argument is deferred to Proposition B.3.

## 5. Applications

To demonstrate the versatility of our results for OCO with movement costs, in this section we consider two applications: OCO with delayed feedback and OCO with time-varying memory. We show that both problems can be reduced to instances of OCO with movement costs, thus allowing us to directly apply Algorithm 3 to obtain near-optimal regret bounds for both problems.

### 5.1. Unconstrained OCO with Delayed Feedback

We first address the problem of unconstrained OCO with delayed feedback. The interaction proceeds as follows: at each round $t \in [T]$, the learner selects a decision $w_t \in \mathcal{W} = \mathbb{R}^n$ while the environment simultaneously chooses a convex loss function $f_t : \mathcal{W} \to \mathbb{R}$, and the learner incurs the loss $f_t(w_t)$. However, unlike the standard OCO setting, the gradient $g_t = \nabla f_t(w_t)$ is only revealed at the end of round $t + d_t$, where $d_t \geq 0$ is an arbitrary delay unknown to the learner. Without loss of generality, we assume $t + d_t \leq T$ for all $t$, ensuring all feedback is received by the end of the game. The learner's performance is measured by the standard dynamic regret:

$$R_T^{\text{del}}(u_{1:T}) \triangleq \sum_{t=1}^{T} \left( f_t(w_t) - f_t(u_t) \right),$$

where the superscript del is used to indicate that the underlying problem setting considered is that of OCO with delayed feedback.

To facilitate our analysis, we introduce the following standard notations for OCO with delayed feedback. Let $o_t \triangleq \{\tau \in \mathbb{Z}_+ : \tau + d_\tau < t\} \subseteq [t-1]$ denote the set of rounds whose gradients have been observed prior to round $t$. Accordingly, let $m_t \triangleq [t-1] \setminus o_t$ denote the set of rounds whose gradients have not yet arrived prior to round $t$. Using these definitions, the feedback protocol is equivalent to receiving the batch of gradients $\{g_\tau\}_{\tau \in o_{t+1} \setminus o_t}$ at the end of round $t$. Finally, we define $\sigma_{\max} \triangleq \max_{t \in [T]} |m_t|$ as the maximum number of outstanding gradients at any step, and $d_{\text{tot}} \triangleq \sum_{t=1}^{T} d_t$ as the total delay.

To utilize Algorithm 3, which is designed for OCO with movement costs, we show a novel connection between delayed feedback and movement penalties. To our knowledge, this connection has not been previously explored in the literature. The following technical lemma bounds the dynamic regret $R_T^{\text{del}}(u_{1:T})$ by a linearized loss constructed from the feedback observed at each round, plus a movement cost rescaled by the number of missing observations.

**Lemma 5.1.** *Assume that $f_1, \ldots, f_T$ are $G$-Lipschitz convex functions. Then*

$$R_T^{\text{del}}(u_{1:T}) \leq \sum_{t=1}^{T} \left\langle \sum_{\tau \in o_{t+1} \setminus o_t} g_\tau, w_t - u_t \right\rangle$$

$$+ G \sum_{t=1}^{T} |m_t| \|w_t - w_{t-1}\| + G P_T \sigma_{\max}.$$

The proof of this lemma is deferred to Appendix C.1. Specifically, Lemma 5.1 decomposes the dynamic regret $R_T^{\text{del}}$ into three components: (i) a linearized regret term driven by the aggregated gradients $\sum_{\tau \in o_{t+1} \setminus o_t} g_\tau$ received at the end of

round $t$; (ii) a movement penalty $G \sum_t |m_t| \|w_t - w_{t-1}\|$, weighted by the count of missing gradients; and (iii) a constant additive term $GP_T \sigma_{\max}$ that is independent of the learner's decisions. This structure suggests a direct reduction from OCO with delayed feedback to OCO with movement costs: we simply treat the aggregated gradients as the instantaneous feedback and set the movement-cost coefficient at round $t$ to $\lambda_t = G|m_t|$. This highlights the necessity of incorporating *time-varying* movement-cost coefficients since $|m_t|$ is naturally fluctuating over time and only revealed at the beginning of round $t$. From an intuitive perspective, a larger number of missing gradients increases the cost for updating the decision while lacking that information about past rounds, making the learner more conservative. Conversely, when less information is missing, the effective movement cost decreases, encouraging faster adaptation as feedback becomes promptly available.

We are now ready to apply our results to OCO with delayed feedback and the complete algorithm is shown in Algorithm 4. In particular, Algorithm 4 serves as a wrapper that implements the reduction to movement costs by running a base learner $\mathcal{A}^{\mathrm{mov}}$, which is an instance of Algorithm 3. At each time step $t \in [T]$, the wrapper adopts the decision $w_t = w_t^{\mathrm{mov}}$ proposed by the base learner. Once $w_t$ is played, the learner observes the batch of newly revealed gradients, aggregated as $h_t = \sum_{\tau \in o_{t+1} \setminus o_t} g_\tau$. Using the count $|o_{t+1} \setminus o_t|$ of these arrivals, the learner calculates the current number of missing gradients, $|m_{t+1}|$. In accordance with Lemma 5.1, we set the movement-cost coefficient for the subsequent update to $\lambda_{t+1} = G|m_{t+1}|$. Finally, the pair $(h_t, \lambda_{t+1})$ is fed to $\mathcal{A}^{\mathrm{mov}}$, which performs the update and outputs the decision for the next round, $w_{t+1}^{\mathrm{mov}}$.

Adapting Theorem 4.1 to this specific choice of gradients and movement coefficients leads to the following dynamic regret bound for OCO with delayed feedback.

**Theorem 5.2.** *Assume that $f_1, \ldots, f_T$ are $G$-Lipschitz convex functions. For any comparator sequence $(u_1, \ldots, u_T) \in \mathcal{W}^T$, Algorithm 4 with any $L \geq G(1 + 3\sigma_{\max})$ and any $\epsilon > 0$ guarantees that*

$$R_T^{\mathrm{del}}(u_{1:T}) = \mathcal{O}\Big( \big(\epsilon \log T + \widetilde{M}_T(\epsilon) + \widetilde{P}_T(\epsilon)\big) L$$

$$+ \sqrt{\big(\widetilde{M}_T(\epsilon) + \widetilde{P}_T(\epsilon)\big) \sum_{t \in [T]} (\|g_t\|^2 + G|\widehat{m}_t|\|g_t\|)\|\widehat{u}_t\|} \Big)$$

$$= \widetilde{\mathcal{O}}\Big( (M + P_T) L + G\sqrt{(M^2 + MP_T)(T + d_{\mathrm{tot}})} \Big).$$

*where we define $\widehat{m}_t = m_{t+d_t}$ and $\widehat{u}_t = u_{t+d_t}$, while $M$, $\widetilde{M}_T(\epsilon)$, and $\widetilde{P}_T(\epsilon)$ are defined as in Theorem 3.1.*

*Proof sketch.* We combine the reduction of Lemma 5.1 with our movement-cost guarantee (Theorem 4.1). First, note

**Algorithm 4** Delays-to-movements Reduction

**Input:** $L > 0, \epsilon > 0$
**Initialize:** Create an instance $\mathcal{A}^{\mathrm{mov}}$ of Algorithm 3 with input $L$ and $\epsilon$. Denote $w_1^{\mathrm{mov}}$ to be the first decision of $\mathcal{A}^{\mathrm{mov}}$.
**for** $t = 1, 2, \ldots$ **do**
    Play $w_t = w_t^{\mathrm{mov}}$
    Observe $h_t = \sum_{\tau \in o_{t+1} \setminus o_t} g_\tau$ and $|o_{t+1} \setminus o_t|$
    Send $(h_t, G|m_{t+1}|)$ to $\mathcal{A}^{\mathrm{mov}}$ and receive $w_{t+1}^{\mathrm{mov}}$

that $\|h_t\| \leq G|o_{t+1} \setminus o_t| \leq G(\sigma_{\max} + 1)$ and that the movement cost scale is $\lambda_{t+1} = G|m_{t+1}| \leq G\sigma_{\max}$, hence the assumption on $L$ satisfies the requirement of Theorem 4.1. Then, using both Lemma 5.1 and Theorem 4.1, we obtain the regret guarantee of the desired form, but where the sum inside the square root is $\sum_{t=1}^T (\|h_t\|^2 + G|m_t|\|h_t\|)\|u_t\|$. After massaging the sum, we can bound it from above by $\sum_{s=1}^T (\|g_s\|^2 + 2G|m_{s+d_s}|\|g_s\|)\|u_{s+d_s}\|$, also thanks to the fact that $s + d_s = t$ for all $s \in o_{t+1} \setminus o_t$. Finally, using $\|g_s\| \leq G$ and $\|u_{s+d_s}\| \leq M$, and applying Lemma C.1 in Appendix C.1 to show that $\sum_{s=1}^T |m_{s+d_s}| = \mathcal{O}(d_{\mathrm{tot}})$, we recover the $d_{\mathrm{tot}}$-dependent worst-case bound as well. $\square$

Theorem 5.2 (whose proof is deferred to Appendix C.1) yields a parameter-free dynamic regret guarantee for delayed feedback on the unconstrained domain $\mathcal{W} = \mathbb{R}^n$, where the impact of delays mainly enters through the $T + d_{\mathrm{tot}}$ factor in the leading term. This can be contrasted with the existing $\widetilde{\mathcal{O}}\big(\sqrt{(1 + P_T)(T + d_{\max}T)}\big)$ dynamic regret guarantee by Wan et al. (2024), who study the problem only over bounded convex domains and achieve a worse $d_{\max}T \gg d_{\mathrm{tot}}$ dependence on delays; their result improves to the desirable $d_{\mathrm{tot}}$ dependence only under an additional "in-order" assumption on delays. Closer in spirit, Van der Hoeven et al. (2022) develop a comparator-adaptive algorithm for unconstrained OCO with delayed feedback and obtain *static* regret bounds featuring a second-order "lag" term that captures the complexity introduced by delays. While their bound exhibits a better adaptivity to gradient norms, our analysis additionally accommodates drifting comparators which come with additional technical challenges.

Note that, in order for the instance $\mathcal{A}^{\mathrm{mov}}$ of Algorithm 3 to provide the desired guarantee, we need $L \geq G(1 + 3\sigma_{\max})$ which would in turn require knowledge of the delay-dependent quantity $\sigma_{\max}$. Nonetheless, Algorithm 3 could adapt to it in a black-box manner as discussed in Remark 4.2.

An immediate corollary of our results is a parameter-free dynamic regret bound for OCO with *both* delayed feedback and movement costs (Pan et al., 2022; Yang et al., 2025). The setting was also studied by (Yang et al., 2025), but with a fixed movement-cost parameter. We denote the movement cost regret under delayed feedback by $R_T^{\mathrm{del}}(u_{1:T}, \lambda_{1:T})$.

**Corollary 5.3.** *Under the same assumptions as Theorem 5.2, suppose that at the end of each round $t$ we feed $\mathcal{A}^{\mathrm{mov}}$ with movement coefficient $\widehat{\lambda}_{t+1} = G|m_{t+1}| + \lambda_{t+1}$. Then for any $L \geq G(1+3\sigma_{\max})+2\lambda_{\max}$ and any $\epsilon > 0$, Algorithm 4 guarantees*

$$R_T^{\mathrm{del}}(u_{1:T}, \lambda_{1:T}) = \mathcal{O}\bigg( \big(\epsilon \log(T) + \widetilde{M}_T(\epsilon) + \widetilde{P}_T(\epsilon)\big)L$$

$$+\sqrt{\big(\widetilde{M}_T(\epsilon)+\widetilde{P}_T(\epsilon)\big)\sum_{t=1}^{T}\big(\|g_t\|^2 + \widehat{\lambda}_{t+d_t}\|g_t\|\big)\|u_{t+d_t}\|}\bigg),$$

*where we define $M$, $\widetilde{M}_T(\epsilon)$, and $\widetilde{P}_T(\epsilon)$ as in Theorem 3.1.*

### 5.2. Unconstrained OCO with Time-varying Memory

We now turn to our second application: unconstrained OCO with time-varying memory. The protocol is defined as follows. The interaction proceeds in $T$ rounds. At each round $t$, the learner selects a decision $w_t \in \mathcal{W} = \mathbb{R}^n$. Simultaneously, the environment chooses a loss function $f_t : \mathcal{W}^{b_t+1} \to \mathbb{R}$ that depends on the current decision and the $b_t$ preceding ones, where $b_t \in \{0, \ldots, t-1\}$ is the memory length. We assume the sequence of memory lengths $\{b_t\}_{t=1}^{T}$ is known to the learner in advance, and we define $B \triangleq \max_{t \in [T]} b_t$. The learner incurs the loss $f_t(w_{t-b_t:t})$, where $w_{t-b_t:t} \triangleq (w_{t-b_t}, \ldots, w_t)$, and subsequently observes the full loss function $f_t$. Note that while observing the full function is standard in the OCO with memory literature (Anava et al., 2015; Zhao et al., 2023), it is a richer form of feedback compared to the gradient feedback considered so far. The learner's goal is to minimize the dynamic regret with respect to any comparator sequence $(u_1, \ldots, u_T) \in \mathcal{W}^T$, defined as follows:

$$R_T^{\mathrm{mem}}(u_{1:T}) \triangleq \sum_{t=1}^{T}\big(f_t(w_{t-b_t:t}) - f_t(u_{t-b_t:t})\big) .$$

Following Anava et al. (2015), we impose structural assumptions on the loss functions. First, we require that the losses are *convex in memory*, meaning that the induced unary function $\widehat{f}_t(w) \triangleq f_t(w, \ldots, w)$ is convex. Second, we assume coordinate-wise Lipschitz continuity and bounded gradients for the unary functions (Anava et al., 2015; Zhao et al., 2023).

**Assumption 5.4** (Coordinate-wise Lipschitzness). *The loss function $f_t$ is coordinate-wise $G$-Lipschitz for some $G \geq 0$, i.e., for all $t \in [T]$ and inputs $x = (x_0, x_1, \ldots, x_{b_t}), y = (y_0, y_1, \ldots, y_{b_t}) \in \mathcal{W}^{b_t+1}$,*

$$|f_t(x_0, \ldots, x_{b_t}) - f_t(y_0, \ldots, y_{b_t})| \leq G\sum_{i=0}^{b_t}\|x_i - y_i\| .$$

**Assumption 5.5** (Bounded gradient norm). *The gradient norm of the unary losses is bounded by $H \geq 0$, i.e., $\|\nabla \widehat{f}_t(w)\| \leq H$ for all $w \in \mathcal{W}$ and $t \in [T]$.*

---

**Algorithm 5** Memory-to-movements Reduction

**Input:** $L > 0$ and $\epsilon > 0$
**Initialize:** Create an instance $\mathcal{A}^{\mathrm{mov}}$ of Algorithm 3 with $L$ and $\epsilon$; denote $w_1^{\mathrm{mov}}$ to be the first decision of $\mathcal{A}^{\mathrm{mov}}$
**for** $t = 1, 2, \ldots$ **do**
  Play $w_t = w_t^{\mathrm{mov}}$
  Observe $f_t : \mathcal{W}^{b_t+1} \to \mathbb{R}$ and $\{s \geq t : s - b_s = t\}$
  Compute $h_t = \nabla \widehat{f}_t(w_t)$
  Set $\xi_{t+1} = \sum_{s=t+1}^{T} \mathbf{1}\{t \geq s - b_s\}(t - (s - b_s) + 1)$
  Send $(h_t, G\xi_{t+1})$ to $\mathcal{A}^{\mathrm{mov}}$ and receive $w_{t+1}^{\mathrm{mov}}$

---

Under Assumption 5.4, a standard analysis allows us to upper bound the memory-dependent loss $f_t(w_{t-b_t:t})$ using the unary loss $\widehat{f}_t(w_t)$ plus a penalty term proportional to the movement cost of the iterates. Applying this decomposition to the comparator's sequence as well yields the following reduction, effectively transforming the problem into an OCO instance with time-varying movement costs.

**Lemma 5.6.** *Suppose $f_1, \ldots, f_T$ are convex in memory and satisfy Assumption 5.4. Then, $R_T^{\mathrm{mem}}(u_{1:T})$ is bounded above by*

$$\sum_{t=1}^{T} \langle h_t, w_t - u_t \rangle + G\sum_{t=2}^{T}\xi_t\|w_t - w_{t-1}\| + GP_T B^2 ,$$

*where $\xi_t = \sum_{s=t}^{T}\mathbf{1}\{t \geq s - b_s + 1\}(t - (s - b_s))$ and $h_t = \nabla \widehat{f}_t(w_t)$ is the unary loss gradient at $w_t$.*

The proof is deferred to Appendix C.2. With this reduction, Algorithm 5 employs Algorithm 3 as a base learner. Specifically, at each round $t$, the algorithm computes the unary gradient $h_t = \nabla \widehat{f}_t(w_t)$ and the effective movement coefficient $\lambda_t = G\xi_t$, which is computable given the memory lengths $\{b_t\}$, and feeds them to the base learner. Combining this reduction with the regret bounds from Theorem 4.1 yields the following parameter-free guarantee for unconstrained OCO with memory.

**Theorem 5.7.** *Suppose $f_1, \ldots, f_T$ are convex in memory and satisfy Assumptions 5.4 and 5.5. For any comparator sequence $(u_1, \ldots, u_T) \in \mathcal{W}^T$, running Algorithm 5 with any $L \geq H + 2GB^2$ and any $\epsilon > 0$ guarantees*

$$R_T^{\mathrm{mem}}(u_{1:T}) = \mathcal{O}\bigg( \big(\epsilon \log T + \widetilde{M}_T(\epsilon) + \widetilde{P}_T(\epsilon)\big)L$$

$$+ \sqrt{M\big(\widetilde{M}_T(\epsilon) + \widetilde{P}_T(\epsilon)\big)\big(H^2 T + GH\sum_{t=1}^{T}b_t^2\big)}\bigg),$$

*where $M$, $\widetilde{M}_T(\epsilon)$, and $\widetilde{P}_T(\epsilon)$ are defined in Theorem 3.1.*

The proof of Theorem 5.7 can be found in Appendix C.2. In terms of related work, Zhao et al. (2023) study dynamic

regret of online learning with fixed-length memory $B \geq 1$ and a constrained domain $\mathcal{W}$ with bounded diameter $D$, and obtain a $\widetilde{\mathcal{O}}\left(\sqrt{D(D + P_T)(B\sqrt{G}H^2 + GHB^2)T}\right)$ parameter-free dynamic regret bound. By contrast, while our bound considers unconstrained domains, Theorem 5.7 shows an improved dependence on memory lengths and Lipschitz constants. For the fixed-memory setting where $b_t = B$ for all $t \in [T]$, the $\Theta(B\sqrt{T})$ dependence on the memory length is known to be minimax optimal for static regret; for instance, see Kumar et al. (2023, Theorem 3.4). Our bound recovers this minimax optimal bound when the memory length $b_t$ is constant over time. Moreover, the same memory dependence extends to dynamic regret: using a standard epoch-based construction—see, e.g., Zhang et al. (2018, Theorem 2)—one can extend the $B\sqrt{T}$ static-regret lower bound to a dynamic-regret lower bound of $\Omega\left(B\sqrt{(1 + P_T)T}\right)$. Thus, up to logarithmic and other problem-dependent constant factors, the dependence of our dynamic regret guarantee on the memory length is minimax optimal.

## 6. Conclusion

In this paper, we provide new dynamic regret guarantees for several natural extensions of the OCO problem. We developed the first comparator-adaptive dynamic regret guarantees for OCO with time-varying movement costs, and we achieve a result which adapts to all problem-dependent quantities of interest. This result also allows us to develop the first comparator-adaptive guarantees in settings with delayed feedback and settings with time-varying memory. All of our algorithms can be efficiently implemented in $\mathcal{O}(d \log(T))$ per-round computation, matching the best-known complexity for obtaining $\sqrt{P_T}$ dependencies in the vanilla OCO setting.

## Acknowledgments

EE and AJ acknowledge the EU Horizon CL4-2022-HUMAN-02 research and innovation action under grant agreement 101120237, project ELIAS (European Lighthouse of AI for Sustainability). HQ gratefully acknowledges the support of Singapore NRF AI Visiting Professorship grant. This work was initiated while HQ was a PhD student at Università degli Studi di Milano.

## Impact Statement

This paper presents work whose goal is to advance the field of Machine Learning. There are many potential societal consequences of our work, none which we feel must be specifically highlighted here.

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

## A. Omitted Details from Section 3

We begin by analyzing the dynamic regret of Algorithm 1 in the following proposition. While this analysis follows a similar structure as the analogous one by Jacobsen & Cutkosky (2022), we require additional care in handling movement costs with time-varying coefficients.

**Proposition A.1.** *Assume that $f_1, \ldots, f_T$ are G-Lipschitz convex functions. Then, for any comparator sequence $u_1, \ldots, u_T \in \mathcal{W}$ and any $\lambda_2, \ldots, \lambda_T \geq 0$, Algorithm 1 with $\eta \leq \frac{1}{G + \lambda_{\max}}$ and $\epsilon_0 > 0$ guarantees*

$$R_T(u_{1:T}, \lambda_{1:T}) \leq \frac{2\|u_T\| \log\left(\frac{\|u_T\|T}{\epsilon_0} + 1\right) + 2\sum_{t=2}^{T}\|u_t - u_{t-1}\| \log\left(\frac{2\|u_t - u_{t-1}\|T^2}{\epsilon_0} + 1\right)}{\eta}$$

$$+ 2\eta \sum_{t=1}^{T}\left(\|g_t\|^2 + \lambda_{t+1}^2\right)\|u_t\| + \frac{1}{\eta T}\sum_{t=1}^{T}\|u_t\| + \epsilon_0\left(G + \lambda_{\max}\right) ,$$

*Proof.* Using Lemma 1 in Jacobsen & Cutkosky (2022), we have

$$R_T(u_{1:T}, \lambda_{1:T}) \leq \sum_{t=1}^{T}\langle g_t, w_t - u_t\rangle + \sum_{t=1}^{T}\lambda_t\|w_t - w_{t-1}\| \qquad \text{(convexity of } f_t\text{)}$$

$$\leq \psi(u_T) + \sum_{t=1}^{T}\varphi_t(u_t) + \sum_{t=2}^{T}\langle\nabla\psi(w_t), u_t - u_{t-1}\rangle$$

$$+ \sum_{t=1}^{T}\left(\langle g_t, w_t - w_{t+1}\rangle - D_\psi(w_{t+1}|w_t) - \varphi_t(w_{t+1})\right) + \sum_{t=1}^{T}\lambda_t\|w_t - w_{t-1}\|$$

$$\text{(Lemma 1 in Jacobsen \& Cutkosky (2022))}$$

$$\overset{(*)}{\leq} \psi(u_T) + \sum_{t=1}^{T}\varphi_t(u_t) + \sum_{t=2}^{T}\langle\nabla\psi(w_t), u_t - u_{t-1}\rangle$$

$$+ \sum_{t=1}^{T}\left(\left(\|g_t\| + \lambda_{t+1}\right)\|w_{t+1} - w_t\| - D_\psi(w_{t+1}|w_t) - \eta\beta_t^2\|w_{t+1}\| - \gamma\|w_{t+1}\|\right) \qquad (2)$$

$$\leq \psi(u_T) + \sum_{t=1}^{T}\varphi_t(u_t) + \sum_{t=2}^{T}\underbrace{\left(\langle\nabla\psi(w_t), u_t - u_{t-1}\rangle - \gamma\|w_t\|\right)}_{\triangleq \mathcal{P}_t}$$

$$+ \sum_{t=1}^{T}\underbrace{\left(\beta_t\|w_{t+1} - w_t\| - D_\psi(w_{t+1}|w_t) - \eta\beta_t^2\|w_{t+1}\|\right)}_{\triangleq \delta_t} , \qquad (3)$$

where we define $\beta_t \triangleq \|g_t\| + \lambda_{t+1}$, $(*)$ expands the definition of $\varphi_t$ and re-indexes $\sum_{t=1}^{T}\lambda_t\|w_t - w_{t-1}\| = \sum_{t=1}^{T}\lambda_{t+1}\|w_t - w_{t+1}\|$ by recalling that we define $w_0 = w_1$, so that $\sum_{t=1}^{T}\lambda_t\|w_t - w_{t-1}\| = \sum_{t=2}^{T}\lambda_t\|w_t - w_{t-1}\| = \sum_{t=1}^{T-1}\lambda_{t+1}\|w_{t+1} - w_t\|$ and recalling that we set $\lambda_{T+1} = 0$ without loss of generality. The last inequality follows by dropping a non-positive term $-\gamma\|w_{T+1}\|$.

Next, we individually bound the two terms $\sum_{t=2}^{T}\mathcal{P}_t$ and $\sum_{t=1}^{T}\delta_t$. We first focus on the sum $\sum_{t=2}^{T}\mathcal{P}_t$. By definition of $\psi(w) = \frac{2}{\eta}\int_0^{\|w\|}\log(x/\alpha + 1)dx$, we have that $\|\nabla\psi(w_t)\| = \frac{2}{\eta}\log(\|w_t\|/\alpha + 1)$, so

$$\sum_{t=2}^{T}\mathcal{P}_t = \sum_{t=2}^{T}\left(\langle\nabla\psi(w_t), u_t - u_{t-1}\rangle - \gamma\|w_t\|\right)$$

$$\leq \sum_{t=2}^{T}\left(\frac{2\log\left(\|w_t\|/\alpha + 1\right)}{\eta}\|u_t - u_{t-1}\| - \gamma\|w_t\|\right)$$

$$\leq \sum_{t=2}^{T}\frac{2\|u_t - u_{t-1}\| \log\left(\frac{2\|u_t - u_{t-1}\|}{\alpha\eta\gamma} + 1\right)}{\eta} , \qquad (4)$$

where the first inequality follows by applying the Cauchy-Schwarz inequality, and the second inequality is due to Lemma D.2 with $a = \alpha$, $b = \eta$, $c = \gamma$, and $d = 2\|u_t - u_{t-1}\|$.

Next, consider the terms $\sum_{t=1}^{T} \delta_t$. By Taylor's theorem, there exists $\widetilde{w}_t$ on the line segment between $w_t$ and $w_{t+1}$ such that

$$D_\psi(w_{t+1}|w_t) = \frac{1}{2}\|w_t - w_{t+1}\|^2_{\nabla^2\psi(\widetilde{w}_t)} \geq \|w_t - w_{t+1}\|^2 \frac{1}{(\|\widetilde{w}_t\| + \alpha)\eta},$$

where $\|v\|^2_{\nabla^2\psi(w)} \triangleq \langle v, \nabla^2\psi(w)v \rangle$ and the inequality holds since $\nabla^2\psi(w) \succeq \frac{2}{\|w\|+\alpha)\eta}I_n$ for any $w \in \mathbb{R}^d$ following Jacobsen & Cutkosky (2022, Proposition 2). Hence, we obtain

$$\sum_{t=1}^{T} \delta_t \leq \sum_{t=1}^{T}\left(\beta_t\|w_t - w_{t+1}\| - \|w_t - w_{t+1}\|^2 \frac{1}{(\|\widetilde{w}_t\| + \alpha)\eta} - \eta\beta_t^2\|w_{t+1}\|\right)$$

$$\leq \sum_{t=1}^{T}\left(\beta_t\|w_t - w_{t+1}\| - \|w_t - w_{t+1}\|^2 \frac{1}{(\|\widetilde{w}_t\| + \alpha)\eta} - \eta\beta_t^2\|\widetilde{w}_t\| + \eta\beta_t^2\|w_{t+1} - \widetilde{w}_t\|\right) \quad \text{(triangle inequality)}$$

$$\leq \sum_{t=1}^{T}\left(\beta_t\|w_t - w_{t+1}\| - \|w_t - w_{t+1}\|^2 \frac{1}{(\|\widetilde{w}_t\| + \alpha)\eta} - \eta\beta_t^2\|\widetilde{w}_t\| + \eta\beta_t^2\|w_{t+1} - w_t\|\right)$$

$$\text{($\widetilde{w}_t$ is on the line segment between $w_t$ and $w_{t+1}$)}$$

$$\overset{(a)}{\leq} \sum_{t=1}^{T}\left(2\beta_t\|w_t - w_{t+1}\| - \|w_t - w_{t+1}\|^2 \frac{1}{(\|\widetilde{w}_t\| + \alpha)\eta} - \eta\beta_t^2\|\widetilde{w}_t\|\right)$$

$$\overset{(b)}{\leq} \sum_{t=1}^{T}\left(\eta\beta_t^2(\alpha + \|\widetilde{w}_t\|) - \eta\beta_t^2\|\widetilde{w}_t\|\right)$$

$$= \eta\alpha\sum_{t=1}^{T}\beta_t^2 , \tag{5}$$

where $(a)$ uses the fact that $\eta \leq \frac{1}{G+\lambda_{\max}} \leq \frac{1}{\beta_t}$ and $(b)$ uses AM-GM inequality. Plugging Equations (4) and (5) into Equation (3) leads to

$$R_T(u_{1:T}, \lambda_{1:T}) \leq \psi(u_T) + \frac{2\sum_{t=2}^{T}\|u_t - u_{t-1}\| \log\left(\frac{2\|u_t - u_{t-1}\|}{\alpha\eta\gamma} + 1\right)}{\eta} + \sum_{t=1}^{T}\varphi_t(u_t) + \alpha\eta\sum_{t=1}^{T}\beta_t^2$$

$$\leq \frac{2\|u_T\| \log\left(\frac{\|u_T\|}{\alpha} + 1\right) + 2\sum_{t=2}^{T}\|u_t - u_{t-1}\| \log\left(\frac{2\|u_t - u_{t-1}\|}{\alpha\eta\gamma} + 1\right)}{\eta}$$

$$+ \sum_{t=1}^{T}\eta\beta_t^2\|u_t\| + \gamma\sum_{t=1}^{T}\|u_t\| + \eta\alpha\sum_{t=1}^{T}\beta_t^2 , \tag{6}$$

where the second inequality holds by the definition of $\varphi_t(w)$ and $\psi(u)$ from Algorithm 1. We proceed by bounding each of the terms in the last line.

By definition of $\beta_t = \|g_t\| + \lambda_{t+1}$, we have that

$$\sum_{t=1}^{T}\eta\beta_t^2\|u_t\| = \sum_{t=1}^{T}\eta\left(\|g_t\| + \lambda_{t+1}\right)^2\|u_t\| \leq 2\sum_{t=1}^{T}\eta\left(\|g_t\|^2 + \lambda_{t+1}^2\right)\|u_t\| . \tag{7}$$

Next, by setting $\gamma = 1/(\eta T)$, we can bound the second summation as

$$\gamma\sum_{t=1}^{T}\|u_t\| = \frac{1}{\eta T}\sum_{t=1}^{T}\|u_t\| \leq \frac{M}{\eta} . \tag{8}$$

Lastly, by the definition of $\alpha = \epsilon_0/T$ and using the assumption that $\eta \leq \frac{1}{G+\lambda_{\max}}$, we can bound the final summation above as

$$\eta\alpha\sum_{t=1}^{T}\beta_t^2 \leq \frac{\epsilon_0}{T\left(G+\lambda_{\max}\right)}\sum_{t=1}^{T}(\|g_t\|+\lambda_{t+1})^2 \leq \frac{\epsilon_0}{T\left(G+\lambda_{\max}\right)}\cdot T\left(G+\lambda_{\max}\right)^2 = \epsilon_0(G+\lambda_{\max})\,, \qquad (9)$$

where the second inequality holds by the facts that $\|g_t\| \leq G$, since the loss functions are $G$-Lipschitz, and $\lambda_{t+1} \leq \lambda_{\max}$, by definition of $\lambda_{\max}$. Plugging Equations (7) to (9) back to Equation (6), we conclude that

$$R_T(u_{1:T},\lambda_{1:T}) \leq \frac{2\|u_T\|\log\left(\frac{\|u_T\|}{\alpha}+1\right) + 2\sum_{t=2}^{T}\|u_t - u_{t-1}\|\log\left(\frac{2\|u_t - u_{t-1}\|}{\alpha\eta\gamma}+1\right)}{\eta}$$
$$+ 2\sum_{t=1}^{T}\eta\left(\|g_t\|^2 + \lambda_{t+1}^2\right)\|u_t\| + \frac{1}{\eta T}\sum_{t=1}^{T}\|u_t\| + \epsilon_0\left(G+\lambda_{\max}\right)$$
$$= \frac{2\|u_T\|\log\left(\frac{\|u_T\|T}{\epsilon_0}+1\right) + 2\sum_{t=2}^{T}\|u_t - u_{t-1}\|\log\left(\frac{2\|u_t - u_{t-1}\|T^2}{\epsilon_0}+1\right)}{\eta}$$
$$+ 2\sum_{t=1}^{T}\eta\left(\|g_t\|^2 + \lambda_{t+1}^2\right)\|u_t\| + \frac{1}{\eta T}\sum_{t=1}^{T}\|u_t\| + \epsilon_0\left(G+\lambda_{\max}\right)\,,$$

where the equality holds by using $\alpha = \epsilon_0/T$ and $\alpha\eta\gamma = \epsilon_0/T^2$. $\qquad\square$

At this point, we may leverage the regret guarantee of Algorithm 1 proved in Proposition A.1 to demonstrate the regret bound for our parameter-free procedure (Algorithm 2). We restate our result (Theorem 3.1) for completeness and provide a complete proof as follows.

**Theorem 3.1.** *Assume that $f_1,\ldots,f_T$ are $G$-Lipschitz convex functions. For any comparator sequence $(u_1,\ldots,u_T) \in \mathcal{W}^T$, Algorithm 2 with any $L \geq G + \lambda_{\max}$ and any $\epsilon > 0$ guarantees that*

$$R_T(u_{1:T},\lambda_{1:T}) = \mathcal{O}\left(\left(\epsilon\log T + \widetilde{M}_T(\epsilon) + \widetilde{P}_T(\epsilon)\right)L + \sqrt{\left(\widetilde{M}_T(\epsilon) + \widetilde{P}_T(\epsilon)\right)\sum_{t=1}^{T}\left(\|g_t\|^2 + \lambda_{t+1}^2\right)\|u_t\|}\right)\,,$$

*where $\widetilde{M}_T(\epsilon) \triangleq M\left(1 + \log\left(\frac{MT}{\epsilon}+1\right)\right)$, $M = \max_t\|u_t\|$, and $\widetilde{P}_T(\epsilon) \triangleq P_T\left(1 + \log\left(\frac{4MT^2}{\epsilon}+1\right)\right)$.*

*Proof.* Recall that Algorithm 2 denotes by $w_t^{\eta_i}$ the iterate chosen by the instance $\mathcal{A}^{\eta_i}$ of the base algorithm (Algorithm 1 to be precise) with learning rate $\eta_i \in \mathcal{S}$ at round $t \in [T]$. Further recall that the decision of Algorithm 2 at each round $t$ ultimately corresponds to $w_t = \sum_{\eta_i\in\mathcal{S}} w_t^{\eta_i}$.

For any $\eta_i \in \mathcal{S}$, we decompose the regret as follows:

$$R_T(u_{1:T},\lambda_{1:T}) \leq \sum_{t=1}^{T}\langle g_t, w_t - u_t\rangle + \sum_{t=1}^{T}\lambda_t\|w_t - w_{t-1}\| \qquad\text{(convexity of $f_t$)}$$
$$= \sum_{t=1}^{T}\left\langle g_t, \sum_{\eta_j\in\mathcal{S}} w_t^{\eta_j} - u_t\right\rangle + \sum_{t=1}^{T}\lambda_t\left\|\sum_{\eta_j\in\mathcal{S}}(w_t^{\eta_j} - w_{t-1}^{\eta_j})\right\|$$
$$\leq \sum_{t=1}^{T}\left(\left\langle g_t, \sum_{\eta_j\in\mathcal{S}} w_t^{\eta_j} - u_t\right\rangle + \lambda_t\sum_{\eta_j\in\mathcal{S}}\|w_t^{\eta_j} - w_{t-1}^{\eta_j}\|\right) \qquad\text{(triangle inequality)}$$
$$= \underbrace{\sum_{t=1}^{T}\langle g_t, w_t^{\eta_i} - u_t\rangle + \sum_{t=1}^{T}\lambda_t\|w_t^{\eta_i} - w_{t-1}^{\eta_i}\|}_{\triangleq R_T^{(i)}(u_{1:T},\lambda_{1:T})} + \underbrace{\sum_{j\neq i}\sum_{t=1}^{T}\langle g_t, w_t^{\eta_j} - \mathbf{0}\rangle + \sum_{t=1}^{T}\lambda_t\|w_t^{\eta_j} - w_{t-1}^{\eta_j}\|}_{\triangleq R_T^{(j)}(\mathbf{0},\lambda_{1:T})} \qquad (10)$$
$$\leq R_T^{(i)}(u_{1:T},\lambda_{1:T}) + (|\mathcal{S}| - 1)\epsilon(G+\lambda_{\max})\,,$$

where the last inequality holds because each instance $\mathcal{A}^{\eta_j}$ of Algorithm 1 with $\epsilon_0 = \epsilon$ and $\eta = \eta_j$ satisfies the conditions of Proposition A.1 since $\eta_i \leq \frac{1}{L} \leq \frac{1}{G+\lambda_{\max}}$ for all $i \in [|\mathcal{S}|]$, hence each algorithm guarantees $R_T^{(j)}(\mathbf{0}, \lambda_{1:T}) \leq \epsilon(G + \lambda_{\max})$ using Proposition A.1. Likewise, for any $\eta_i \in \mathcal{S}$, applying the result of Proposition A.1 again on the regret $R_T^{(i)}(u_{1:T}, \lambda_{1:T})$ of $\mathcal{A}^{\eta_i}$ gives us

$$R_T(u_{1:T}, \lambda_{1:T}) \leq \frac{2\|u_T\| \log\left(\frac{\|u_T\|T}{\epsilon} + 1\right) + 2\sum_{t=2}^{T}\|u_t - u_{t-1}\| \log\left(\frac{2\|u_t - u_{t-1}\|T^2}{\epsilon} + 1\right)}{\eta_i}$$

$$+ 2\sum_{t=1}^{T}\eta_i\left(\|g_t\|^2 + \lambda_{t+1}^2\right)\|u_t\| + \frac{1}{\eta_i T}\sum_{t=1}^{T}\|u_t\| + |\mathcal{S}|\epsilon\left(G + \lambda_{\max}\right)$$

$$\leq \frac{2M\left(\log\left(\frac{MT}{\epsilon} + 1\right) + 1\right) + 2\sum_{t=2}^{T}\|u_t - u_{t-1}\| \log\left(\frac{2\|u_t - u_{t-1}\|T^2}{\epsilon} + 1\right)}{\eta_i}$$

$$+ 2\sum_{t=1}^{T}\eta_i\left(\|g_t\|^2 + \lambda_{t+1}^2\right)\|u_t\| + |\mathcal{S}|\epsilon\left(G + \lambda_{\max}\right) ,$$

where the second inequality uses $M = \max_t \|u_t\|$. Then, using Lemma D.1, we know that there exists an $\eta_i \in \mathcal{S}$ such that

$$R_T(u_{1:T}, \lambda_{1:T}) \leq \mathcal{O}\left(\sqrt{\left[M\left(\log\left(\frac{MT}{\epsilon} + 1\right) + 1\right) + \sum_{t=2}^{T}\|u_t - u_{t-1}\| \log\left(\frac{2\|u_t - u_{t-1}\|T^2}{\epsilon} + 1\right)\right]\sum_{t=1}^{T}\left(\|g_t\|^2 + \lambda_{t+1}^2\right)\|u_t\|}\right.$$

$$+ \frac{M\left(\log\left(\frac{MT}{\epsilon} + 1\right) + 1\right) + \sum_{t=2}^{T}\|u_t - u_{t-1}\| \log\left(\frac{2\|u_t - u_{t-1}\|T^2}{\epsilon} + 1\right)}{\eta_{\max}}$$

$$+ \eta_{\min}\sum_{t=1}^{T}\left(\|g_t\|^2 + \lambda_{t+1}^2\right)\|u_t\| + |\mathcal{S}|\epsilon(G + \lambda_{\max})\right)$$

$$= \mathcal{O}\left(\sqrt{\left(\widetilde{M}_T(\epsilon) + \widetilde{P}_T(\epsilon)\right)\sum_{t=1}^{T}\left(\|g_t\|^2 + \lambda_{t+1}^2\right)\|u_t\|} + \left(|\mathcal{S}|\epsilon + \widetilde{M}_T(\epsilon) + \widetilde{P}_T(\epsilon)\right)L\right)$$

where $\widetilde{M}_T(\epsilon) = M\left(1 + \log\left(\frac{MT}{\epsilon} + 1\right)\right)$, $\widetilde{P}_T(\epsilon) = P_T\left(1 + \log\left(\frac{4MT^2}{\epsilon} + 1\right)\right)$, and the last step holds by observing that $|\mathcal{S}| = \mathcal{O}(\log T)$ and using $\|u_t - u_{t-1}\| \leq \|u_t\| + \|u_{t-1}\| \leq 2M$, $\eta_{\max} = \frac{1}{L}$, and

$$\eta_{\min} = \frac{1}{L\sqrt{T}} \leq \sqrt{\frac{1}{\sum_{t=1}^{T}(\|g_t\|^2 + \lambda_{t+1}^2)}} \leq \sqrt{\frac{M}{\sum_{t=1}^{T}(\|g_t\|^2 + \lambda_{t+1}^2)\|u_t\|}}.$$

$\square$

## B. Omitted Details from Section 4

Here we present the details omitted from Section 4. We begin by stating two useful lemmas that will be central to the regret analysis of Algorithm 3. These lemmas establish key properties of Algorithm 3 that will be used in the proof. For ease of exposition, let $N$ be the number of blocks and define $I_\tau = \{t_\tau, \ldots, t_{\tau+1} - 1\}$ as the $\tau$-th epoch defined by the execution of Algorithm 3, for each $\tau \in [N]$.

The first result illustrates how to translate the dependence of the regret of Algorithm 3 on the norm of the accumulated gradient $\widetilde{g}_\tau$ over each epoch $\tau$ and the movement-cost scale $\widetilde{\lambda}_{\tau+1}$ incurred by updating the decision at the end of it, into a dependence on the original gradients $g_t$ and movement coefficients $\lambda_t$ of rounds $t \in I_\tau$ in the same epoch.

**Lemma B.1.** *For every $\tau \in [N]$, Algorithm 3 guarantees that*

$$\|\widetilde{g}_\tau\|^2 + \widetilde{\lambda}_{\tau+1}^2 \leq \sum_{t \in I_\tau}\left(2\|g_t\|^2 + 4\lambda_t\|g_t\|\right) .$$

*Proof.* Fix any $\tau \in [N]$. We first focus on bounding $\widetilde{\lambda}_{\tau+1}^2$. According to the updates in Algorithm 3, we know that $\widetilde{\lambda}_{\tau+1} = \lambda_{t_{\tau+1}} < \|\widetilde{g}_\tau\| = \|\sum_{t \in I_\tau} g_t\|$ and thus

$$\widetilde{\lambda}_{\tau+1}^2 \leq \left\|\sum_{t \in I_\tau} g_t\right\|^2 = \left(\sum_{t \in I_\tau} \|g_t\|^2 + 2 \sum_{s,t \in I_\tau : s < t} \langle g_s, g_t \rangle\right). \tag{11}$$

Notice that this trivially holds for $\tau = N$: we are free to define $\widetilde{\lambda}_{N+1} = 0$ without loss of generality in the analysis, so we have $\widetilde{\lambda}_{N+1} = 0 \leq \|\sum_{t \in I_N} g_t\|$ in the first step. Now notice that the second term in the right-hand side of Equation (11) satisfies:

$$\sum_{s,t \in I_\tau : s < t} \langle g_s, g_t \rangle = \sum_{t \in I_\tau} \left\langle \sum_{s=t_\tau}^{t-1} g_s, g_t \right\rangle \leq \sum_{t \in I_\tau} \left\|\sum_{s=t_\tau}^{t-1} g_s\right\| \|g_t\| \leq \sum_{t \in I_\tau} \lambda_t \|g_t\|, \tag{12}$$

where the second inequality holds due to the fact that $\|\sum_{s=t_\tau}^{t-1} g_s\| \leq \lambda_t$ when $t \in \{t_\tau + 1, \ldots, t_{\tau+1} - 1\}$. Hence, we finally obtain that

$$\widetilde{\lambda}_{\tau+1}^2 \leq \sum_{t \in I_\tau} \left(\|g_t\|^2 + 2\lambda_t \|g_t\|\right). \tag{13}$$

Next we consider bounding the term $\|\widetilde{g}_\tau\|^2$. We similarly have that

$$\|\widetilde{g}_\tau\|^2 = \left\|\sum_{t \in I_\tau} g_t\right\|^2 \leq \sum_{t \in I_\tau} \left(\|g_t\|^2 + 2\lambda_t \|g_t\|\right), \tag{14}$$

where the inequality follows by the same reasoning as in Equations (11) and (12). Combining Equation (14) and Equation (13) finishes the proof. □

The second useful result presented in this section shows how to move from the original regret $R_T(u_{1:T}, \lambda_{1:T})$ to a linearized regret defined over epochs instead of rounds.

**Lemma B.2.** *Assume that $f_1, \ldots, f_T$ are $G$-Lipschitz convex functions. For any comparator sequence $(u_t)_{t \in [T]} \in \mathcal{W}^T$, Algorithm 3 with any base algorithm $\mathcal{A}$ guarantees that*

$$R_T(u_{1:T}, \lambda_{1:T}) \leq \left(G + 2\lambda_{\max}\right) P_T + \underbrace{\sum_{\tau=1}^N \langle \widetilde{g}_\tau, \widetilde{w}_\tau - \widetilde{u}_\tau \rangle + \widetilde{\lambda}_\tau \|\widetilde{w}_\tau - \widetilde{w}_{\tau-1}\|}_{\widetilde{R}_N(\widetilde{u}_{1:N}, \widetilde{\lambda}_{1:N})}$$

*where, for each epoch $\tau \in [N]$: $\widetilde{u}_\tau \in \{u_t : t \in I_\tau\}$, $\widetilde{w}_\tau$ denotes the action played by $\mathcal{A}$, $\widetilde{\lambda}_\tau = \lambda_{t_\tau}$ represents the movement cost at the start of the epoch, and $\widetilde{g}_\tau \triangleq \sum_{t \in I_\tau} g_t$.*

*Proof.* By convexity of $f_1, \ldots, f_T$ and update rules in Algorithm 3, we know that

$$R_T(u_{1:T}, \lambda_{1:T}) = \sum_{t=1}^T \left(f_t(w_t) - f_t(u_t)\right) + \lambda_t \|w_t - w_{t-1}\| \tag{15}$$

$$\leq \sum_{t=1}^T \langle g_t, w_t - u_t \rangle + \lambda_t \|w_t - w_{t-1}\| \quad \text{(convexity of } f_t)$$

$$= \sum_{t=1}^T \langle g_t, w_t - u_t \rangle + \sum_{\tau=1}^N \widetilde{\lambda}_\tau \|\widetilde{w}_\tau - \widetilde{w}_{\tau-1}\|, \tag{16}$$

where the last line follows from the fact that $w_t = w_{t-1}$ except when transitioning between epochs, and defining $\widetilde{w}_0 = \widetilde{w}_1$ and $\widetilde{\lambda}_1 = 0$ for notational convenience. Focusing on the first summation, and letting $\widetilde{u}_\tau$ be an arbitrary comparator from $\{u_t : t \in I_\tau\}$, we have

$$\sum_{t=1}^T \langle g_t, w_t - u_t \rangle = \sum_{\tau=1}^N \sum_{t \in I_\tau} \langle g_t, w_t - \widetilde{u}_\tau \rangle + \sum_{\tau=1}^N \sum_{t \in I_\tau} \langle g_t, \widetilde{u}_\tau - u_t \rangle$$

$$
= \sum_{\tau=1}^{N} \sum_{t \in I_\tau} \langle g_t, \widetilde{w}_\tau - \widetilde{u}_\tau \rangle + \sum_{\tau=1}^{N} \sum_{t \in I_\tau} \langle g_t, \widetilde{u}_\tau - u_t \rangle
$$

$$
= \sum_{\tau=1}^{N} \langle \widetilde{g}_\tau, \widetilde{w}_\tau - \widetilde{u}_\tau \rangle + \sum_{\tau=1}^{N} \sum_{t \in I_\tau} \langle g_t, \widetilde{u}_\tau - u_t \rangle, \tag{17}
$$

where the second equality holds by choice of $w_t = \widetilde{w}_\tau$ when $t \in I_\tau$, and the third equality is due to the definition of $\widetilde{g}_\tau$. For each $\tau \in [N]$, we recall that $u_{t_{\tau+1}-1}$ is the last comparator of epoch $\tau$ by definition of $t_{\tau+1}$. Then observe that

$$
\sum_{\tau=1}^{N} \sum_{t \in I_\tau} \langle g_t, u_{t_{\tau+1}-1} - u_t \rangle = \sum_{\tau=1}^{N} \sum_{t \in I_\tau} \Big\langle g_t, \sum_{s \in I_\tau : s > t} (u_s - u_{s-1}) \Big\rangle
$$

$$
= \sum_{\tau=1}^{N} \sum_{s \in I_\tau : s > t_\tau} \Big\langle \sum_{t=t_\tau}^{s-1} g_t, u_s - u_{s-1} \Big\rangle
$$

$$
\leq \sum_{\tau=1}^{N} \sum_{s \in I_\tau : s > t_\tau} \Big\| \sum_{t=t_\tau}^{s-1} g_t \Big\| \| u_s - u_{s-1} \|
$$

$$
\leq \sum_{\tau=1}^{N} \sum_{s \in I_\tau : s > t_\tau} \lambda_s \| u_s - u_{s-1} \|
$$

$$
= \sum_{t=2}^{T} \lambda_t \| u_t - u_{t-1} \|, \tag{18}
$$

where in the second inequality, we use that, for any $s \in I_\tau$ with $s > t_\tau$, the epoch $\tau$ has not terminated before time $s-1$, and hence $\big\| \sum_{t=t_\tau}^{s-1} g_t \big\| \leq \lambda_s$ due to how epochs are determined by Algorithm 3. Using this, we can bound $\sum_{\tau=1}^{N} \sum_{t \in I_\tau} \langle g_t, \widetilde{u}_\tau - u_t \rangle$ as follows:

$$
\sum_{\tau=1}^{N} \sum_{t \in I_\tau} \langle g_t, \widetilde{u}_\tau - u_t \rangle = \sum_{\tau=1}^{N} \Big\langle \sum_{t \in I_\tau} g_t, \widetilde{u}_\tau - u_{t_{\tau+1}-1} \Big\rangle + \sum_{\tau=1}^{N} \sum_{t \in I_\tau} \langle g_t, u_{t_{\tau+1}-1} - u_t \rangle
$$

$$
\leq \sum_{\tau=1}^{N} \Big\langle \sum_{t \in I_\tau} g_t, \widetilde{u}_\tau - u_{t_{\tau+1}-1} \Big\rangle + \sum_{t=2}^{T} \lambda_t \| u_t - u_{t-1} \| \qquad \text{(Equation (18))}
$$

$$
\leq \sum_{\tau=1}^{N} \Big\| \sum_{t \in I_\tau} g_t \Big\| \cdot \| \widetilde{u}_\tau - u_{t_{\tau+1}-1} \| + \lambda_{\max} P_T
$$

$$
= \sum_{\tau=1}^{N} \| \widetilde{g}_\tau \| \cdot \| \widetilde{u}_\tau - u_{t_{\tau+1}-1} \| + \lambda_{\max} P_T . \tag{19}
$$

Observe that, by the triangle inequality,

$$
\| \widetilde{g}_\tau \| \leq \Big\| \sum_{t \in I_\tau} g_t - g_{t_{\tau+1}-1} \Big\| + G \leq \max_{t \in I_\tau} \lambda_t + G ,
$$

where the last step follows from the same observation on how epoch $I_\tau$ is defined by the execution of Algorithm 3. Plugging the above inequality into Equation (19) yields

$$
\sum_{\tau=1}^{N} \sum_{t \in I_\tau} \langle g_t, \widetilde{u}_\tau - u_t \rangle \leq \sum_{\tau=1}^{N} \| \widetilde{g}_\tau \| \cdot \| \widetilde{u}_\tau - u_{t_{\tau+1}-1} \| + \lambda_{\max} P_T
$$

$$
\leq \sum_{\tau=1}^{N} \Big( G + \max_{t \in I_\tau} \lambda_t \Big) \| \widetilde{u}_\tau - u_{t_{\tau+1}-1} \| + \lambda_{\max} P_T
$$

$$\leq \left(G + \max_{t\in[T]} \lambda_t\right) \sum_{\tau=1}^{N} \|\widetilde{u}_\tau - u_{t_{\tau+1}-1}\| + \lambda_{\max} P_T$$

$$\leq \left(G + 2\lambda_{\max}\right) P_T \; ,$$

where the last inequality uses the fact that $\widetilde{u}_\tau \in \{u_t : t \in I_\tau\}$. Combining the above with Equation (17) and plugging back into the full regret bound in Equation (16) concludes the proof. $\qquad\square$

We now have all the ingredients to prove the main result of Section 4, stated in Theorem 4.1. For completeness, we restate the theorem and provide its proof in what follows.

**Theorem 4.1.** *Assume that $f_1, \ldots, f_T$ are $G$-Lipschitz convex functions. For any comparator sequence $(u_1, \ldots, u_T) \in \mathcal{W}^T$, Algorithm 3 with any $L \geq G + 2\lambda_{\max}$ and any $\epsilon > 0$ guarantees that*

$$R_T(u_{1:T}, \lambda_{1:T}) = \mathcal{O}\left(\left(\epsilon \log T + \widetilde{M}_T(\epsilon) + \widetilde{P}_T(\epsilon)\right)L + \sqrt{\left(\widetilde{M}_T(\epsilon) + \widetilde{P}_T(\epsilon)\right) \sum_{t=1}^{T}\left(\|g_t\|^2 + \lambda_t\|g_t\|\right)\|u_t\|}\right),$$

*where $M$, $\widetilde{M}_T(\epsilon)$ and $\widetilde{P}_T(\epsilon)$ are defined as in Theorem 3.1.*

*Proof.* Using Lemma B.2, we have for all $(u_t)_{t\in[T]}$ and $(\widetilde{u}_\tau)_{\tau\in[N]}$ where $u_\tau \in \{u_t : t \in I_\tau\}$,

$$R_T(u_{1:T}, \lambda_{1:T}) \leq \sum_{\tau=1}^{N}\langle \widetilde{g}_\tau, \widetilde{w}_\tau - \widetilde{u}_\tau\rangle + \sum_{t=1}^{T}\lambda_t\|w_t - w_{t-1}\| + \left(G + 2\lambda_{\max}\right)P_T \tag{20}$$

$$= \underbrace{\sum_{\tau=1}^{N}\langle \widetilde{g}_\tau, \widetilde{w}_\tau - \widetilde{u}_\tau\rangle + \sum_{\tau=1}^{N}\widetilde{\lambda}_\tau\|\widetilde{w}_\tau - \widetilde{w}_{\tau-1}\|}_{\triangleq \widetilde{R}_N(\widetilde{u}_{1:N}, \widetilde{\lambda}_{1:N})} + \left(G + 2\lambda_{\max}\right)P_T \; .$$

Note that for every $\tau \in [N]$, $\widetilde{\lambda}_{\tau+1} \leq \lambda_{\max}$ and

$$\|\widetilde{g}_\tau\| \leq \left\|\sum_{t\in I_\tau} g_t - g_{t_{\tau+1}-1}\right\| + \|g_{t_{\tau+1}-1}\| \leq \max_{t\in I_\tau}\lambda_t + G \leq \lambda_{\max} + G$$

according to the update of Algorithm 3, where the first step follows by triangle inequality. Applying Theorem 3.1 with $L = G + 2\lambda_{\max} \geq \max_\tau\left(\|\widetilde{g}_\tau\| + \widetilde{\lambda}_{\tau+1}\right)$ we can immediately show that[1]

$$\widetilde{R}_N(\widetilde{u}_{1:N}, \widetilde{\lambda}_{1:N}) = \mathcal{O}\left(\sqrt{\left(\widetilde{M}_N(\epsilon) + \widetilde{P}_N(\epsilon)\right)\sum_{\tau=1}^{N}\left(\|\widetilde{g}_\tau\|^2 + \widetilde{\lambda}_{\tau+1}^2\right)\|\widetilde{u}_\tau\|} + \left(|\mathcal{S}|\epsilon + \widetilde{M}_N(\epsilon) + \widetilde{P}_N(\epsilon)\right)\left(G + \lambda_{\max}\right)\right), \tag{21}$$

where

$$\widetilde{M}_N(\epsilon) = M'\left(\log\left(\frac{M'T}{\epsilon} + 1\right) + 1\right) \leq \widetilde{M}_T(\epsilon) \tag{22}$$

with $M' \triangleq \max_{\tau\in[N]}\|\widetilde{u}_\tau\| \leq M$, and

$$\widetilde{P}_N(\epsilon) = \sum_{\tau=2}^{N}\|\widetilde{u}_\tau - \widetilde{u}_{\tau-1}\|\left(1 + \log\left(\frac{4MT^2}{\epsilon} + 1\right)\right)$$

---

[1]Note that in the analysis of Theorem 3.1, the last epoch may not have actually closed at time $T$, but it is easily seen that we may take the last epoch to be $I_N = \{t_N, \ldots, T\}$, which still satisfies $\widetilde{\lambda}_{N+1} \leq \|\sum_{t\in I_N} g_t\| \leq \widetilde{\lambda}_N + G$ since we may take $\widetilde{\lambda}_{N+1} = 0$ without loss of generality in the analysis. These details could be made explicit by adding a condition which closes the epoch and sets $\widetilde{\lambda}_{N+1} = 0$ on round $T$, though we opt for the more concise algorithm description for ease of exposition.

$$\leq \sum_{t=2}^{T} \|u_t - u_{t-1}\| \left(1 + \log\left(\frac{4MT^2}{\epsilon} + 1\right)\right) = \widetilde{P}_T(\epsilon), \tag{23}$$

via triangle inequality and the fact that $(\widetilde{u}_\tau)_{\tau=1}^{N}$ defines a subsequence of the original comparator sequence $(u_t)_{t=1}^{T}$.[2] Moreover, applying Lemma B.1 and picking $\widetilde{u}_\tau \in \arg\min_{u \in \{u_t : t \in I_\tau\}} \|u\|$ for each $\tau \in [N]$, we have that

$$\sum_{\tau=1}^{N} (\|\widetilde{g}_\tau\|^2 + \widetilde{\lambda}_{\tau+1}^2) \|\widetilde{u}_\tau\| \leq \sum_{\tau=1}^{N} \|\widetilde{u}_\tau\| \sum_{t \in I_\tau} \left(2\|g_t\|^2 + 4\lambda_t \|g_t\|\right) \leq \sum_{t=1}^{T} \left(2\|g_t\|^2 + 4\lambda_t \|g_t\|\right) \|u_t\|. \tag{24}$$

Combining Equations (20) to (24) and recalling that $|\mathcal{S}| = \mathcal{O}(\log T)$ finish the proof. $\qquad\square$

The following proposition compares the first-order regret bound with the second-order bound. It shows that the first-order guarantee proven in Theorem 4.1 is never worse than Theorem 3.1, yet can be much sharper in regimes where the gradients are small.

**Proposition B.3.** *The regret bound shown in Theorem 4.1 is no worse than that in Theorem 3.1, up to constant factors.*

*Proof.* We compare the following two regret bounds. By Theorem 3.1,

$$R_T(u_{1:T}, \lambda_{1:T}) = \mathcal{O}\left(\left(\epsilon \log T + \widetilde{M}_T(\epsilon) + \widetilde{P}_T(\epsilon)\right)L + \sqrt{\left(\widetilde{M}_T(\epsilon) + \widetilde{P}_T(\epsilon)\right) \sum_{t=1}^{T} \left(\|g_t\|^2 + \lambda_{t+1}^2\right)\|u_t\|}\right),$$

whereas by Theorem 4.1,

$$R_T(u_{1:T}, \lambda_{1:T}) = \mathcal{O}\left(\left(\epsilon \log T + \widetilde{M}_T(\epsilon) + \widetilde{P}_T(\epsilon)\right)L + \sqrt{\left(\widetilde{M}_T(\epsilon) + \widetilde{P}_T(\epsilon)\right) \sum_{t=1}^{T} \left(\|g_t\|^2 + \lambda_t\|g_t\|\right)\|u_t\|}\right).$$

The first terms in the two bounds are identical. Hence it suffices to compare the second terms. By AM-GM inequality, we have

$$\lambda_t \|g_t\| \leq \frac{1}{2}\lambda_t^2 + \frac{1}{2}\|g_t\|^2.$$

Therefore,

$$\sqrt{\left(\widetilde{M}_T(\epsilon) + \widetilde{P}_T(\epsilon)\right) \sum_{t=1}^{T} \left(\|g_t\|^2 + \lambda_t\|g_t\|\right)\|u_t\|} \leq \mathcal{O}\left(\sqrt{\left(\widetilde{M}_T(\epsilon) + \widetilde{P}_T(\epsilon)\right) \sum_{t=1}^{T} \left(\|g_t\|^2 + \lambda_t^2\right)\|u_t\|}\right).$$

It remains to relate the term involving $\lambda_t^2$ to the one involving $\lambda_{t+1}^2$. By the triangle inequality,

$$\|u_t\| \leq \|u_{t-1}\| + \|u_t - u_{t-1}\|.$$

Hence,

$$\begin{aligned}
\sum_{t=1}^{T} \lambda_t^2 \|u_t\| &\leq \sum_{t=1}^{T} \lambda_t^2 \|u_{t-1}\| + \sum_{t=1}^{T} \lambda_t^2 \|u_t - u_{t-1}\| \\
&\leq \sum_{t=1}^{T} \lambda_t^2 \|u_{t-1}\| + \lambda_{\max}^2 P_T && (u_0 = u_1 \text{ and } P_T = \sum_{t=2}^{T}\|u_t - u_{t-1}\|) \\
&\leq \sum_{t=1}^{T} \lambda_{t+1}^2 \|u_t\| + \lambda_{\max}^2 (M + P_T),
\end{aligned}$$

---

[2] Note that the grid of learning rates in Algorithm 2 should still be defined in terms of $T \geq N$ since $N$ is unknown *a priori*, so the terms $\widetilde{P}_N(\epsilon)$ and $\widetilde{M}_N(\epsilon)$ naturally exhibit logarithmic penalties which depend on $T$ rather than $N$ in Equations (22) and (23)

where the last inequality follows by re-indexing and using $\lambda_{T+1} = 0$ and bounding $\|u_0\| = \|u_1\| \leq M$. Therefore,

$$\sqrt{\left(\widetilde{M}_T(\epsilon) + \widetilde{P}_T(\epsilon)\right) \sum_{t=1}^{T} \left(\|g_t\|^2 + \lambda_t \|g_t\|\right) \|u_t\|}$$

$$\leq \mathcal{O}\left(\sqrt{\left(\widetilde{M}_T(\epsilon) + \widetilde{P}_T(\epsilon)\right) \sum_{t=1}^{T} \left(\|g_t\|^2 + \lambda_{t+1}^2\right) \|u_t\| + \lambda_{\max}\sqrt{\left(\widetilde{M}_T(\epsilon) + \widetilde{P}_T(\epsilon)\right)(M + P_T)}}\right).$$

Since $M + P_T \leq \widetilde{M}(\epsilon) + \widetilde{P}_T(\epsilon)$, we further get

$$\sqrt{\left(\widetilde{M}_T(\epsilon) + \widetilde{P}_T(\epsilon)\right) \sum_{t=1}^{T} \left(\|g_t\|^2 + \lambda_t \|g_t\|\right) \|u_t\|}$$

$$\leq \mathcal{O}\left(\sqrt{\left(\widetilde{M}_T(\epsilon) + \widetilde{P}_T(\epsilon)\right) \sum_{t=1}^{T} \left(\|g_t\|^2 + \lambda_{t+1}^2\right) \|u_t\| + \lambda_{\max}\left(\widetilde{M}_T(\epsilon) + \widetilde{P}_T(\epsilon)\right)}\right).$$

Finally, since $\lambda_{\max} \leq L$, we know that $\lambda_{\max}\left(\widetilde{M}_T(\epsilon) + \widetilde{P}_T(\epsilon)\right) = \mathcal{O}\left(\left(\epsilon \log T + \widetilde{M}_T(\epsilon) + \widetilde{P}_T(\epsilon)\right)L\right)$. Combining the bounds above leads to the conclusion. $\qquad\square$

## C. Omitted Details from Section 5

### C.1. Unconstrained OCO with Delayed Feedback

In this section, we first introduce useful lemmas that will be crucial in the regret analysis of Algorithm 4. The following lemma (initially stated as Lemma 5.1 in Section 5.1) provides a reduction from OCO with delayed feedback to OCO with movement costs as described in Section 5.1.

**Lemma 5.1.** *Assume that $f_1, \ldots, f_T$ are $G$-Lipschitz convex functions. Then*

$$R_T^{\mathrm{del}}(u_{1:T}) \leq \sum_{t=1}^{T} \left\langle \sum_{\tau \in o_{t+1} \setminus o_t} g_\tau, w_t - u_t \right\rangle + G\sum_{t=1}^{T} |m_t| \|w_t - w_{t-1}\| + GP_T \sigma_{\max}.$$

*Proof.* We decompose the regret as follows:

$$R_T^{\mathrm{del}}(u_{1:T}) = \sum_{t=1}^{T} \left(f_t(w_t) - f_t(u_t)\right)$$

$$\leq \sum_{t=1}^{T} \langle g_t, w_t - u_t \rangle \qquad\qquad\qquad \text{(convexity of } f_t)$$

$$= \sum_{t=1}^{T} \langle g_t, w_t - u_t \rangle - \underbrace{\sum_{t=1}^{T} \left\langle \sum_{\tau \in o_{t+1} \setminus o_t} g_\tau, w_t - u_t \right\rangle}_{\clubsuit} + \clubsuit. \qquad (25)$$

Define $z_{t+1} = z_t + \sum_{\tau \in o_{t+1} \setminus o_t} g_\tau - g_t$ for any $t \in [T]$ and $z_1 = \mathbf{0}$. Equivalently, we have $z_t = \sum_{\tau \in o_t} g_\tau - \sum_{\tau=1}^{t-1} g_\tau = -\sum_{\tau \in m_t} g_\tau$. Without loss of generality, assume that $t + d_t \leq T$ for all $t \in [T]$ so that all feedback is received by the end of the game. This is innocuous because feedback arriving after round $T$ cannot affect any learner's decisions within the $T$-round horizon. Then $o_{T+1} = [T]$, and hence $z_{T+1} = \mathbf{0}$. Therefore, we have

$$\sum_{t=1}^{T} \left\langle g_t - \sum_{\tau \in o_{t+1} \setminus o_t} g_\tau, w_t \right\rangle = \sum_{t=1}^{T} \langle z_t - z_{t+1}, w_t \rangle$$

$$= \sum_{t=1}^{T-1} \langle z_{t+1}, w_{t+1} - w_t \rangle \qquad \text{(since } z_{T+1} = z_1 = \mathbf{0})$$

$$= \sum_{t=1}^{T-1} \left\langle \sum_{\tau \in m_{t+1}} g_\tau, w_t - w_{t+1} \right\rangle. \tag{26}$$

Similarly, we obtain

$$\sum_{t=1}^{T} \left\langle g_t - \sum_{\tau \in o_{t+1} \setminus o_t} g_\tau, u_t \right\rangle = \sum_{t=1}^{T-1} \left\langle \sum_{\tau \in m_{t+1}} g_\tau, u_t - u_{t+1} \right\rangle. \tag{27}$$

Then, we know that

$$\sum_{t=1}^{T} \langle g_t, w_t - u_t \rangle - \clubsuit = \sum_{t=1}^{T} \left\langle g_t - \sum_{\tau \in o_{t+1} \setminus o_t} g_\tau, w_t - u_t \right\rangle$$

$$= \sum_{t=1}^{T-1} \left\langle \sum_{\tau \in m_{t+1}} g_\tau, w_t - w_{t+1} \right\rangle + \sum_{t=1}^{T-1} \left\langle \sum_{\tau \in m_{t+1}} g_\tau, u_{t+1} - u_t \right\rangle \quad \text{(by Equations (26) and (27))}$$

$$\leq G \sum_{t=1}^{T-1} |m_{t+1}| \|w_{t+1} - w_t\| + G \sum_{t=1}^{T-1} |m_{t+1}| \|u_{t+1} - u_t\| \qquad \text{(Cauchy–Schwarz and } \|g_t\| \leq G)$$

$$\leq G \sum_{t=1}^{T-1} |m_{t+1}| \|w_{t+1} - w_t\| + G \sigma_{\max} P_T .$$

Here the last inequality follows from $|m_{t+1}| \leq \sigma_{\max}$ and $\sum_{t=1}^{T-1} \|u_{t+1} - u_t\| = \sum_{t=2}^{T} \|u_t - u_{t-1}\| = P_T$. Combining the above inequality and Equation (25) and using the convention $w_0 = w_1$, we finally obtain that

$$R_T^{\text{del}}(u_{1:T}) \leq \sum_{t=1}^{T} \left\langle \sum_{\tau \in o_{t+1} \setminus o_t} g_\tau, w_t - u_t \right\rangle + G \sum_{t=1}^{T} |m_t| \cdot \|w_t - w_{t-1}\| + G P_T \sigma_{\max} . \qquad \square$$

The following lemma quantifies the relationship between $m_t$, $o_t$, and $d_t$. This result is a property of the delay sequence and the respective delay-dependent parameters, regardless of other parts of the online learning problem. Its proof is an extension of similar arguments from the proof of Theorem E.1 in Qiu et al. (2025).

**Lemma C.1.** *For any sequence of delays, $\sum_{t=1}^{T} |m_t| \cdot |o_{t+1} \setminus o_t| \leq 2d_{\text{tot}}$.*

*Proof.* Direct calculations show that

$$\sum_{t=1}^{T} |m_t| \cdot |o_{t+1} \setminus o_t| = \sum_{t=1}^{T} |m_t|(1 + |m_t| - |m_{t+1}|) \qquad (|o_{t+1} \setminus o_t| = |m_t| + 1 - |m_{t+1}|)$$

$$= \sum_{t=1}^{T} |m_t| + |m_1|^2 - |m_T| \cdot |m_{T+1}| + \sum_{t=2}^{T} (|m_t| - |m_{t-1}|)|m_t|$$

$$= \sum_{t=1}^{T} |m_t| + \sum_{t=2}^{T} (|m_t| - |m_{t-1}|)|m_t| \qquad (|m_1| = |m_{T+1}| = 0)$$

$$\leq 2 \sum_{t=1}^{T} |m_t| \qquad (|m_t| \leq |m_{t-1}| + 1)$$

$$\leq 2d_{\text{tot}} .$$

The last inequality follows since each gradient $g_\tau$ can be outstanding in at most $d_\tau$ rounds, so $\sum_{t=1}^{T} |m_t| \leq \sum_{\tau=1}^{T} d_\tau = d_{\text{tot}}$. $\square$

The above two technical lemmas we proved in this section enable the proof of our unconstrained parameter-free dynamic regret bound for the setting of OCO with delayed feedback, achieved by our Algorithm 4. This is one of the two main implications of our results for OCO with time-varying movement costs. In the following, we provide the formal proof of Theorem 5.2, which we restate here for completeness.

**Theorem 5.2.** *Assume that $f_1, \ldots, f_T$ are $G$-Lipschitz convex functions. For any comparator sequence $(u_1, \ldots, u_T) \in \mathcal{W}^T$, Algorithm 4 with any $L \geq G(1 + 3\sigma_{\max})$ and any $\epsilon > 0$ guarantees that*

$$R_T^{\mathrm{del}}(u_{1:T}) = \mathcal{O}\Big(\big(\epsilon \log T + \widetilde{M}_T(\epsilon) + \widetilde{P}_T(\epsilon)\big)L + \sqrt{\big(\widetilde{M}_T(\epsilon) + \widetilde{P}_T(\epsilon)\big)\sum_{t=1}^{T}(\|g_t\|^2 + G|\widehat{m}_t|\|g_t\|)\|\widehat{u}_t\|}\Big)$$

$$= \widetilde{\mathcal{O}}\Big((M + P_T)L + G\sqrt{(M^2 + MP_T)(T + d_{\mathrm{tot}})}\Big).$$

*where we define $\widehat{m}_t = m_{t+d_t}$ and $\widehat{u}_t = u_{t+d_t}$, while $M$, $\widetilde{M}_T(\epsilon)$, and $\widetilde{P}_T(\epsilon)$ are defined as in Theorem 3.1.*

*Proof.* Recall that $h_t = \sum_{\tau \in o_{t+1} \setminus o_t} g_\tau$ and $\lambda_{t+1} = G|m_{t+1}|$ for each $t \in [T]$, as defined by Algorithm 4 when reducing to an instance of time-varying movement costs. Using Lemma 5.1, we have

$$R_T^{\mathrm{del}}(u_{1:T}) \leq \sum_{t=1}^{T}\Big\langle \sum_{\tau \in o_{t+1} \setminus o_t} g_\tau, w_t - u_t \Big\rangle + G\sum_{t=1}^{T}|m_t| \cdot \|w_t - w_{t-1}\| + GP_T\sigma_{\max} \, .$$

At each time step $t \in [T]$, when we run Algorithm 4 with $L \geq G(1 + 3\sigma_{\max})$, an instance $\mathcal{A}^{\mathrm{mov}}$ of Algorithm 3 incurs linear loss $\big\langle \sum_{\tau \in o_{t+1} \setminus o_t} g_\tau, w_t \big\rangle$ plus movement cost $G|m_t|\|w_t - w_{t-1}\|$. By the triangle inequality and the $G$-Lipschitzness of the loss functions, together with the fact that $|o_{t+1} \setminus o_t| \leq |m_t| + 1$, we can show that

$$\|h_t\| = \Big\| \sum_{\tau \in o_{t+1} \setminus o_t} g_\tau \Big\| \leq \sum_{\tau \in o_{t+1} \setminus o_t} \|g_\tau\| \leq G|o_{t+1} \setminus o_t| \leq G(|m_t| + 1) \leq G(\sigma_{\max} + 1) \, , \tag{28}$$

while $\lambda_{t+1} = G|m_{t+1}| \leq G\sigma_{\max}$ directly follows by definition of $\sigma_{\max}$. Hence, $L \geq G(1 + 3\sigma_{\max})$ satisfies the condition in Theorem 4.1, which we can in turn use to obtain that

$$R_T^{\mathrm{del}}(u_{1:T}) = \mathcal{O}\left((|\mathcal{S}|\epsilon + \widetilde{M}_T(\epsilon) + \widetilde{P}_T(\epsilon))L + \sqrt{\big(\widetilde{M}_T(\epsilon) + \widetilde{P}_T(\epsilon)\big)\sum_{t=1}^{T}(\|h_t\|^2 + G|m_t| \cdot \|h_t\|)\|u_t\|}\right) \, .$$

Let us now focus on the sum within the square root. Fix any $t \in [T]$. By definition of $h_t$, each term can be rewritten as

$$\|h_t\|^2 + G|m_t| \cdot \|h_t\| \leq \Big(\sum_{\tau \in o_{t+1} \setminus o_t} \|g_\tau\|\Big)^2 + G|m_t| \sum_{\tau \in o_{t+1} \setminus o_t} \|g_\tau\|$$

$$= \Big(\sum_{\tau \in o_{t+1} \setminus o_t} \|g_\tau\|\Big)^2 + G \sum_{\tau \in o_{t+1} \setminus o_t} \|g_\tau\||m_{\tau + d_\tau}| \, ,$$

where we used the fact that $\tau + d_\tau = t$ for all $\tau \in o_{t+1} \setminus o_t$ at the last equality. Moreover, we have that

$$\Big(\sum_{\tau \in o_{t+1} \setminus o_t} \|g_\tau\|\Big)^2 = \sum_{\tau \in o_{t+1} \setminus o_t} \|g_\tau\|^2 + \sum_{\tau \in o_{t+1} \setminus o_t} \|g_\tau\| \sum_{\substack{s \in o_{t+1} \setminus o_t \\ s \neq \tau}} \|g_s\|$$

$$\leq \sum_{\tau \in o_{t+1} \setminus o_t} \|g_\tau\|^2 + G\big(|o_{t+1} \setminus o_t| - 1\big) \sum_{\tau \in o_{t+1} \setminus o_t} \|g_\tau\|$$

$$\leq \sum_{\tau \in o_{t+1} \setminus o_t} \|g_\tau\|^2 + G|m_t| \sum_{\tau \in o_{t+1} \setminus o_t} \|g_\tau\|$$

$$= \sum_{\tau \in o_{t+1} \backslash o_t} \|g_\tau\|^2 + G \sum_{\tau \in o_{t+1} \backslash o_t} \|g_\tau\| |m_{\tau+d_\tau}| ,$$

where the first inequality holds by $G$-Lipschitzness of the loss functions, the second one by observing that $|o_{t+1} \backslash o_t| \leq |m_t| + 1$, and the last equality by the previously observed fact that $\tau + d_\tau = t$. Then, considering the sum over all rounds $t \in [T]$ yields

$$\sum_{t=1}^{T} (\|h_t\|^2 + G|m_t| \cdot \|h_t\|) \|u_t\| \leq \sum_{t=1}^{T} \|u_t\| \left( \sum_{\tau \in o_{t+1} \backslash o_t} \|g_\tau\|^2 + 2G \sum_{\tau \in o_{t+1} \backslash o_t} \|g_\tau\| |m_{\tau+d_\tau}| \right)$$

$$= \sum_{t=1}^{T} \sum_{\tau \in o_{t+1} \backslash o_t} (\|g_\tau\|^2 + 2G|m_{\tau+d_\tau}| \cdot \|g_\tau\|) \|u_{\tau+d_\tau}\|$$

$$= \sum_{t=1}^{T} (\|g_t\|^2 + 2G|m_{t+d_t}| \cdot \|g_t\|) \|u_{t+d_t}\| .$$

Plugging this back into the regret guarantee obtained so far shows that

$$R_T^{\text{del}}(u_{1:T}) = \mathcal{O} \left( (|\mathcal{S}|\epsilon + \widetilde{M}_T(\epsilon) + \widetilde{P}_T(\epsilon))L + \sqrt{(\widetilde{M}_T(\epsilon) + \widetilde{P}_T(\epsilon)) \sum_{t=1}^{T} (\|g_t\|^2 + G|m_{t+d_t}| \cdot \|g_t\|) \|u_{t+d_t}\|} \right) .$$

By further bounding the sum from the leading term under square root, we can additionally show that

$$\sum_{t=1}^{T} (\|g_t\|^2 + G|m_{t+d_t}| \cdot \|g_t\|) \|u_{t+d_t}\| \leq G^2 M \left( T + 2 \sum_{t=1}^{T} |m_{t+d_t}| \right)$$

$$= G^2 M \left( T + 2 \sum_{t=1}^{T} |m_t| \cdot |o_{t+1} \backslash o_t| \right)$$

$$\leq G^2 M (T + 4d_{\text{tot}}) ,$$

using Lemma C.1 at the last step. This allows us to show that the above regret guarantee additionally recovers the worst-case regret bound depending on the total delay:

$$R_T^{\text{del}}(u_{1:T}) = \mathcal{O} \left( (|\mathcal{S}|\epsilon + \widetilde{M}_T(\epsilon) + \widetilde{P}_T(\epsilon))L + G\sqrt{M(\widetilde{M}_T(\epsilon) + \widetilde{P}_T(\epsilon))(T + d_{\text{tot}})} \right) .$$

Finally, recalling that $|\mathcal{S}| = \mathcal{O}(\log T)$ concludes the proof. $\qquad\square$

### C.2. Unconstrained OCO with Time-varying Memory

In this section, we first introduce a useful lemma (already stated as Lemma 5.6 in Section 5.2) that constitutes a fundamental component in the regret analysis of Algorithm 5. Similarly to existing works, the lemma provides a reduction from OCO with time-varying memory to OCO with time-varying movement costs. The main difference in what we demonstrate is that we can handle arbitrary sequences of memory lengths, which can fluctuate over time, rather than limiting the applicability of our results to a uniform memory length $B$.

**Lemma 5.6.** *Suppose $f_1, \ldots, f_T$ are convex in memory and satisfy Assumption 5.4. Then, $R_T^{\text{mem}}(u_{1:T})$ is bounded above by*

$$\sum_{t=1}^{T} \langle h_t, w_t - u_t \rangle + G \sum_{t=2}^{T} \xi_t \|w_t - w_{t-1}\| + G P_T B^2 ,$$

*where $\xi_t = \sum_{s=t}^{T} \mathbf{1}\{t \geq s - b_s + 1\}(t - (s - b_s))$ and $h_t = \nabla \widehat{f}_t(w_t)$ is the unary loss gradient at $w_t$.*

*Proof.* For any $t \in [T]$, since $f_t$ satisfies Assumption 5.4 and $\widehat{f}_t(w_t) = f_t(w_t, \ldots, w_t)$ with $b_t + 1$ arguments, we have

$$
\left| f_t\left(w_{t-b_t:t}\right) - \widehat{f}_t\left(w_t\right) \right| = \left| f_t\left(w_{t-b_t:t}\right) - f_t(w_t, \ldots, w_t) \right|
$$

$$
\leq G \sum_{i=0}^{b_t} \|w_t - w_{t-i}\|
$$

$$
= G \sum_{i=1}^{b_t} \|w_t - w_{t-i}\|
$$

$$
\leq G \sum_{i=1}^{b_t} \sum_{j=0}^{i-1} \|w_{t-j} - w_{t-j-1}\|
$$

$$
= G \sum_{i=1}^{b_t} (b_t - i + 1) \|w_{t-i+1} - w_{t-i}\|,
$$

where the second inequality follows from applying the triangle inequality to each term $\|w_t - w_{t-i}\|$. Taking a summation over $t \in [T]$, we obtain

$$
\sum_{t=1}^{T} \left| f_t\left(w_{t-b_t:t}\right) - \widehat{f}_t\left(w_t\right) \right| \leq G \sum_{t=1}^{T} \sum_{i=1}^{b_t} (b_t - i + 1) \|w_{t-i+1} - w_{t-i}\|
$$

$$
= G \sum_{t=2}^{T} \left( \sum_{s=t}^{T} \mathbf{1}\{t \geq s - b_s + 1\}(t - (s - b_s)) \right) \|w_t - w_{t-1}\|
$$

$$
= G \sum_{t=2}^{T} \xi_t \|w_t - w_{t-1}\|, \tag{29}
$$

where we use the fact that $w_1 = w_0$, together with the definition

$$
\xi_t \triangleq \sum_{s=t}^{T} \mathbf{1}\{t \geq s - b_s + 1\}(t - (s - b_s)).
$$

An analogous result can be similarly shown to hold for the comparator sequence. Therefore, we can bound $R_T^{\mathrm{mem}}(u_{1:T})$ as follows:

$$
R_T^{\mathrm{mem}}(u_{1:T}) = \sum_{t=1}^{T} \left( \widehat{f}_t(w_t) - \widehat{f}_t(u_t) \right) + \sum_{t=1}^{T} \left( f_t\left(w_{t-b_t:t}\right) - \widehat{f}_t(w_t) \right) - \sum_{t=1}^{T} \left( f_t\left(u_{t-b_t:t}\right) - \widehat{f}_t(u_t) \right)
$$

$$
\leq \sum_{t=1}^{T} \langle h_t, w_t - u_t \rangle + \sum_{t=1}^{T} \left| f_t\left(w_{t-b_t:t}\right) - \widehat{f}_t\left(w_t\right) \right| + \sum_{t=1}^{T} \left| f_t\left(u_{t-b_t:t}\right) - \widehat{f}_t\left(u_t\right) \right|
$$

$$
\leq \sum_{t=1}^{T} \langle h_t, w_t - u_t \rangle + G \sum_{t=2}^{T} \xi_t \|w_t - w_{t-1}\| + G \sum_{t=2}^{T} \xi_t \|u_t - u_{t-1}\|
$$

$$
\leq \sum_{t=1}^{T} \langle h_t, w_t - u_t \rangle + G \sum_{t=2}^{T} \xi_t \|w_t - w_{t-1}\| + G P_T B^2,
$$

where the first inequality uses the convexity of $\widehat{f}_t$ and $h_t = \nabla \widehat{f}_t(w_t)$, the second inequality essentially uses Equation (29), and the last inequality uses the uniform bound $\xi_t \leq B^2$ together with $P_T = \sum_{t=2}^{T} \|u_t - u_{t-1}\|$. Here we use the conventions $u_0 = u_1$ when extending the variation terms to start from $t = 1$. In particular, this last inequality can be proved in the following way: by denoting as $[\cdot]_+ \triangleq \max\{\cdot, 0\}$ the positive part of any given real number, we have

$$
\xi_t = \sum_{s=t}^{T} \mathbf{1}\{t \geq s - b_s + 1\}(t - (s - b_s))
$$

$$= \sum_{s=t}^{T} \big[ t - (s - b_s) \big]_+ \tag{30}$$

$$\leq \sum_{s=t}^{T} \big[ B - (s - t) \big]_+ \qquad \text{(using } b_s \leq B)$$

$$= \sum_{d=0}^{\min\{B-1,\, T-t\}} (B - d)$$

$$\leq \sum_{d=0}^{B-1} (B - d)$$

$$= \frac{B(B+1)}{2}$$

$$\leq B^2 , \tag{31}$$

where the first inequality uses $b_s \leq B$ for all $s$, and the last inequality holds for any integer $B \geq 1$. The case $B = 0$ is trivial since then $\xi_t = 0$. $\qquad \square$

For completeness, we restate Theorem 5.7, the second of our two main applications of the results for OCO with time-varying movement costs derived in Sections 3 and 4, and provide its proof.

**Theorem 5.7.** *Suppose $f_1, \ldots, f_T$ are convex in memory and satisfy Assumptions 5.4 and 5.5. For any comparator sequence $(u_1, \ldots, u_T) \in \mathcal{W}^T$, running Algorithm 5 with any $L \geq H + 2GB^2$ and any $\epsilon > 0$ guarantees*

$$R_T^{\mathrm{mem}}(u_{1:T}) = \mathcal{O}\left( \big( \epsilon \log T + \widetilde{M}_T(\epsilon) + \widetilde{P}_T(\epsilon) \big) L + \sqrt{ M \big( \widetilde{M}_T(\epsilon) + \widetilde{P}_T(\epsilon) \big) \Big( H^2 T + GH \sum_{t=1}^{T} b_t^2 \Big) } \right),$$

*where $M$, $\widetilde{M}_T(\epsilon)$, and $\widetilde{P}_T(\epsilon)$ are defined in Theorem 3.1.*

*Proof.* Using Lemma 5.6, we begin by observing that

$$R_T^{\mathrm{mem}}(u_{1:T}) \leq \sum_{t=1}^{T} \langle h_t, w_t - u_t \rangle + G \sum_{t=2}^{T} \xi_t \|w_t - w_{t-1}\| + G P_T B^2 .$$

At each time step $t \in [T]$, when we run Algorithm 5 with $L \geq H + 2GB^2$, the instance $\mathcal{A}^{\mathrm{mov}}$ of Algorithm 3 incurs a linear loss $\langle h_t, w_t \rangle$ plus a movement penalty of $G \xi_t \|w_t - w_{t-1}\|$. Observe that these two costs satisfy $\|h_t\| = \|\nabla \widehat{f}(w_t)\| \leq H$ by definition of $h_t = \nabla \widehat{f}(w_t)$ in Algorithm 5 together with Assumption 5.5, and $G \xi_t \leq GB^2$ by Equation (31). Hence, $L \geq H + 2GB^2$ satisfies the condition in Theorem 4.1. Applying Theorem 4.1, and absorbing the additive term $G P_T B^2$ into the $L \widetilde{P}_T(\epsilon)$ term since $L \geq GB^2$ and $P_T \leq \widetilde{P}_T(\epsilon)$, we obtain

$$R_T^{\mathrm{mem}}(u_{1:T}) \leq \mathcal{O}\left( \big( |\mathcal{S}|\epsilon + \widetilde{M}_T(\epsilon) + \widetilde{P}_T(\epsilon) \big) L + \sqrt{ \big( \widetilde{M}_T(\epsilon) + \widetilde{P}_T(\epsilon) \big) \sum_{t=1}^{T} \big( \|h_t\|^2 + G \xi_t \|h_t\| \big) \|u_t\| } \right)$$

$$\leq \mathcal{O}\left( \big( |\mathcal{S}|\epsilon + \widetilde{M}_T(\epsilon) + \widetilde{P}_T(\epsilon) \big) L + \sqrt{ M \big( \widetilde{M}_T(\epsilon) + \widetilde{P}_T(\epsilon) \big) \Big( H^2 T + GH \sum_{t=1}^{T} \xi_t \Big) } \right)$$

$$\leq \mathcal{O}\left( \big( |\mathcal{S}|\epsilon + \widetilde{M}_T(\epsilon) + \widetilde{P}_T(\epsilon) \big) L + \sqrt{ M \big( \widetilde{M}_T(\epsilon) + \widetilde{P}_T(\epsilon) \big) \Big( H^2 T + GH \sum_{t=1}^{T} b_t^2 \Big) } \right) . \tag{32}$$

The second inequality uses $\|h_t\| \leq H$ and $\|u_t\| \leq M$. The last inequality follows from $\sum_{t=1}^{T} \xi_t \leq \sum_{t=1}^{T} b_t^2$, which we prove next. Since $\xi_1 = 0$ and $b_1 = 0$, it suffices to bound $\sum_{t=2}^{T} \xi_t$. Using Equation (30), we have

$$\sum_{t=2}^{T} \xi_t = \sum_{t=2}^{T} \sum_{s=t}^{T} \big[ b_s - (s - t) \big]_+ \qquad \text{(using Equation (30))}$$

$$= \sum_{s=2}^{T} \sum_{t=2}^{s} [b_s - (s - t)]_+$$

$$= \sum_{s=2}^{T} \sum_{d=0}^{s-2} [b_s - d]_+$$

$$\leq \sum_{s=2}^{T} \sum_{d=0}^{b_s - 1} (b_s - d)$$

$$= \sum_{s=2}^{T} \frac{b_s(b_s + 1)}{2}$$

$$\leq \sum_{s=2}^{T} b_s^2 ,$$

where $[\cdot]_+ \triangleq \max\{\cdot, 0\}$ denotes the positive part of any given real number. Finally, observing that $|\mathcal{S}| = \mathcal{O}(\log T)$ concludes the proof. $\qquad\square$

## D. Auxiliary Lemmas

In this section, we provide some auxiliary technical results that will be helpful throughout the paper. The first lemma is a crucial component for the analysis of Algorithm 3, essentially showing that there exists a learning rate value in the exponential grid that achieves a close-to-optimal tuning.

**Lemma D.1.** *Let* $0 < \eta_{\min} \leq \eta_{\max}$ *and* $\mathcal{S}_0 = \{\eta_i = 2^i \eta_{\min} \wedge \eta_{\max} : i = 0, 1, \dots\}$. *Then for any* $P \geq 0$ *and* $V \geq 0$, *there exists an* $\eta \in \mathcal{S}_0$ *such that*

$$\frac{P}{\eta} + \eta V \leq 3\sqrt{PV} + \frac{P}{\eta_{\max}} + \eta_{\min} V .$$

*Proof.* Let $\eta^* = \sqrt{P/V}$ be the optimal tuning of $\eta$ minimizing $\frac{P}{\eta} + \eta V$. Suppose that there is $\eta_i \in \mathcal{S}_0$ such that $\eta_i \leq \eta^* \leq \eta_{i+1} = 2\eta_i$. Then choosing $\eta = \eta_i$ yields

$$\frac{P}{\eta} + \eta V = \frac{P}{\eta_i} + \eta_i V \leq \frac{2P}{\eta^*} + \eta^* V = 3\sqrt{PV} .$$

If $\eta^* \leq \eta_{\min}$, then choosing $\eta = \eta_{\min} = \eta_0 \in \mathcal{S}_0$ guarantees

$$\frac{P}{\eta} + \eta V \leq \frac{P}{\eta^*} + \eta_{\min} V = \sqrt{PV} + \eta_{\min} V ,$$

and otherwise, if $\eta^* \geq \eta_{\max}$, then choosing $\eta = \eta_{\max}$ yields

$$\frac{P}{\eta} + \eta V \leq \frac{P}{\eta_{\max}} + \eta^* V = \frac{P}{\eta_{\max}} + \sqrt{PV} .$$

Combining these three cases proves the result:

$$\min_{\eta \in \mathcal{S}_0} \left\{ \frac{P}{\eta} + \eta V \right\} \leq \sqrt{PV} + \max\left\{ 2\sqrt{PV}, \frac{P}{\eta_{\max}}, \eta_{\min} V \right\} \leq 3\sqrt{PV} + \frac{P}{\eta_{\max}} + \eta_{\min} V . \qquad\square$$

The second technical result is a generalization of a similar inequality used in the proof of Proposition 1 in Jacobsen & Cutkosky (2022) for the analysis of their Centered Mirror Descent algorithm. In our case, this inequality is adopted within the regret analysis of Algorithm 1.

**Lemma D.2.** *Given any* $a, b, c > 0$ *and* $d \geq 0$, *it holds that* $\sup_{x \geq 0}\left\{ \frac{d}{b} \log(\frac{x}{a} + 1) - cx \right\} \leq \frac{d}{b} \log\left(\frac{d}{abc} + 1\right)$.

*Proof.* If $d = 0$, then the left-hand side equals $\sup_{x \geq 0}(-cx) = 0$, while the right-hand side is also $0$. Hence the result holds. Now assume $d > 0$, let $f(x) \triangleq \frac{d}{b} \log\left(\frac{x}{a} + 1\right) - cx$. Direct calculation shows that $f'(\frac{d}{bc} - a) = 0$. Since $f''(x) = -\frac{d}{b(x+a)^2} \leq 0$, we consider two cases:

**Case 1:** $\frac{d}{bc} - a \leq 0$. Then, due to concavity, the supremum over $x \geq 0$ occurs at the boundary $x = 0$. Thus, $\sup_{x \geq 0} f(x) = f(0) = 0 \leq \frac{d}{b} \log\left(\frac{d}{abc} + 1\right)$ holds trivially.

**Case 2:** $\frac{d}{bc} - a > 0$. Then the maximum of $f(x)$ is attained at $x = \frac{d}{bc} - a$. Substituting $x = \frac{d}{bc} - a$ into $f(x)$ leads to:

$$\sup_{x \geq 0} f(x) = \frac{d}{b} \log\left(\frac{\frac{d}{bc} - a}{a} + 1\right) - c\left(\frac{d}{bc} - a\right)$$

$$= \frac{d}{b} \log\left(\frac{d}{abc}\right) - \frac{d}{b} + ac = \frac{d}{b}\left(\log \alpha - 1 + \frac{1}{\alpha}\right),$$

where we define $\alpha \triangleq \frac{d}{abc} \geq 1$. To prove the lemma, it suffices to show $f(x^*) \leq \frac{d}{b} \log(\alpha + 1)$, which is equivalent to:

$$\log(\alpha) - 1 + \frac{1}{\alpha} \leq \log(\alpha + 1) \iff \frac{1}{\alpha} - 1 \leq \log\left(1 + \frac{1}{\alpha}\right).$$

Let $z = 1/\alpha$. Since $\alpha > 1$, we have $z \in (0, 1)$. The inequality $z - 1 \leq \log(1 + z)$ holds for all $z > 0$ (as $h(z) = \log(1 + z) - z + 1$ is decreasing for $z \in (0, 1)$ and $h(1) > 0$). Thus, the bound holds. $\square$

