# OpenReview forum: "Parameter-free Dynamic Regret: Time-varying Movement Costs, Delayed Feedback, and Memory"
_ICML.cc/2026/Conference — ICML 2026 regular_

### Official Review · Reviewer_5dcM · 2026-03-12

**Soundness:** 3
**Presentation:** 3
**Significance:** 2
**Originality:** 2
**Overall Recommendation:** 5
**Confidence:** 3

**Summary:**

This paper shows a bound on the dynamic regret for online optimization with convex loss functions and quadratic movement costs. The movement costs are weighted with $\lambda_{t}\geq 0$. The algorithm analysed is a variant of Mirror descent with an additional regularizer $\varphi_t$ to trade off the movement costs and the potential gain for moving into the negative gradient direction (measured in therms of the gradient norm $||g_t||$).
The authors show that their results can be applied to (a) learning with delayed feedback in an unconstraint domain and (b) OCO with memory.

**Compliance With Llm Reviewing Policy:**

Affirmed.

**Final Justification:**

%%%%%%%%%%%%% after rebuttal %%%%%%%%%%%%%%%%
Since my questions were addressed, I decided to raise my score.

**Key Questions For Authors:**

Q1: (p5, second column, line 242 ff) [...] improving the movment costs from $\sum_{t=1}^T \lambda_{t}^2$ to $\sum_{t=1}^T \lambda_{t} || g_t||$ [...].  To the best of my understanding, this is not necessarily an improvment, e.g., if $\lambda_t < 1$ and $||g_t|| > 1$. On the same line: Is it possible to have an algorithm achieving both bounds (Theorem 3.1 and 4.1) simultaneously?

Q2: Are the bounds in Theorem 3.1 and 4.1 tight? Are there any known lower bounds (besides the well known lower bounds for dynamic regret)?

**Limitations:**

This is theoretical work, thus no negative societal impact.

**Strengths And Weaknesses:**

Strengths:
- I find the application examples interesting.
- The paper is well written and clear.

Weaknesess:
- the results/techniques of Theorem 3.1. and 4.1 are not too surprising and mostly incremental improvments. This is fine for me, since I think the main strength of this paper lies in the application examples.
- as far as I can see, the guarantees of Theorem 3.1. and 4.1 can not be obtained simultaneously.
- the limitations could be better discussed. E.g.: Is the bound tight or is this still unknown? Is it possible to improve on the polylog factor?

---

> ### Author Rebuttal · Authors · 2026-03-30
>
> Thank you for the positive review and for your careful reading of our
> paper. We first address a stated weakness of the paper:
>
> > the results/techniques of Theorem 3.1. and 4.1 are not too surprising
> > and mostly incremental improvments
>
> While our approach shares similarities with existing mirror-descent
> based approaches, we emphasize that there are no approaches in prior
> works that are capable of obtaining dynamic regret guarantees for OCO
> with movement costs in the unconstrained setting. In light of this
> distinction, we believe the contribution is better understood as going
> beyond an incremental extension of prior work. Please see our response
> to reviewer YCoU [here](https://openreview.net/forum?id=xnj5CyzVVW&noteId=V53AWVHQwI) for a more detailed discussion on
> this point.
>
> In what follows, we address each of the specific questions.
>
> ### Answers to Questions
>
> **Q1:** improving the movement costs from
> $\sum_{t=1}^T \lambda_t^2 \text { to } \sum_{t=1}^T \lambda_t\left\\|g_t\right\\|$,
> Is it possible to have an algorithm achieving both bounds (Theorem 3.1
> and 4.1) simultaneously?
>
> **A1:** In fact, first-order dependence can always be upper bounded in terms of
> the second-order one by AM-GM inequality:
> $\sum_t \\|g_t\\|^2+\\|g_t\\|\lambda_t\le O(\sum_t\\|g_t\\|^2+\lambda_t^2)$.
> However, the converse is not true, and the gap between these two can be
> very significant.
>
> A convenient illustrative example where the second-order bound is
> significantly worse than the first-order can be seen in our results for
> delayed feedback. In particular, in our reduction for delayed feedback,
> $\lambda_t=G|m_t|$ where $|m_t|$ is the number of missing observations
> at the beginning of $t$. In this case, the first-order dependencies scale
> as
> $\sqrt{\sum_t \\|g_t\\|\lambda_t}\le G \sqrt{\sum_t |m_t|}\le G\sqrt{d_{tot}}$.
> This yields an overall rate of
> $\sqrt{\sum_t \\|g_t\\|^2 + \lambda_t\\|g_t\\|}\le G\sqrt{T+d_{tot}}$, which
> is known to be minimax optimal with respect to $T$ and $d_{tot}$.
> Conversely, the second-order dependence would scale with
> $G\sqrt{\sum_t |m_t|^2}$, a bound that can easily become much worse than
> $G\sqrt{T}$ when $d_{tot}=\Theta(T)$. For instance, if all $g_t$'s where
> $t\leq \sqrt{T}$ are delayed for $\sqrt{T}$ rounds while the remaining ones have no delay, the second-order
> term $G\sqrt{\sum_t|m_t|^2}$ becomes $O(GT^{3/4})$, while the
> first-order bound remains $O(G\sqrt{T})$. We can add a remark discussing
> this illustrative example for the next revision of the paper.
>
>
> **Q2:** Are the bounds in Theorem 3.1 and 4.1 tight? Are there any
> known lower bounds (besides the well known lower bounds for dynamic
> regret)? Is it possible to improve on the polylog factor?
>
> **A2:** While we do not have a formal lower bound for our setting,
> our results match several known minimax optimal bounds. In particular,
> it is known that there is an unavoidable penalty of
> $\Omega(\sqrt{G\lambda(1+P_T)  T})$ for OCO with fixed movement cost
> $\lambda$ (see Theorem 5 in [3]), which matches our result in the worst
> case (i.e., where $\lambda_t=\lambda$ for all $t\in[T]$). For OCO
> without movement costs, it is also known that there is an unavoidable
> penalty of $\Omega(G\sqrt{(1+P_T)T})$ in the worst case for dynamic
> regret in OCO. Taken together, our results capture both of these
> penalties in the worst case, but can be significantly smaller in
> practice due to being adaptive to the individual $\\|g_t\\|_2$ and
> $\lambda_t$.
>
> As for whether there are potential improvements on polylog factor, it is
> known that the "cost" associated with adapting to the comparator-norm
> $\\|u\\|$ in the static regret setting is an additional
> $\sqrt{\log(\\|u\\|\sqrt{T}/\epsilon+1)}$ multiplicative factor (see,
> e.g., Theorem 7 in [2]). Likewise, our dynamic regret bounds exhibit a
> time-varying form of this penalty,
> $O(\sqrt{\log(M T/\epsilon+1)})$ with $M = \\max_t \\|u_t\\|$, which is unsurprising
> since our bound adapts to each of the comparator norms $\\|u_t\\|$
> without prior knowledge. We note that a lower bound showing this penalty
> has not been formally established in the literature, but it would be one
> natural generalization in the dynamic regret setting of the price of adaptivity
> to $\\|u\\|$ from static regret.
>
> **References:**
>
> [1] McMahan, B., & Streeter, M.. No-regret algorithms for
> unconstrained online convex optimization. NeurIPS, 2012.
>
> [2] Zhang, Z., Cutkosky, A., and Paschalidis, Y. Optimal comparator
> adaptive online learning with switching cost. NeurIPS, 2022.

---

> > ### Author Rebuttal · Reviewer_5dcM · 2026-04-02
> >
> > My questions have been adequately addressed. Thus I decided to increase my score.

---

### Official Review · Reviewer_1Aes · 2026-03-13

**Soundness:** 3
**Presentation:** 3
**Significance:** 3
**Originality:** 3
**Overall Recommendation:** 5
**Confidence:** 2

**Summary:**

This paper studies the unconstrained online convex optimization (OCO) setting in which one wants to minimize dynamic regret against evolving baselines, and where there are time-varying costs associated to (drastically) changing decision points. Their main contribution is to give and analyze a parameter-free algorithm, for which they can show a favorable regret bound in terms of problem parameters. Finally, they show how their settings can also capture the settings of OCO with delayed feedback (which is novel) and OCO with time-varying memory, extending prior works to the unconstrained setting with strong regret bounds.

**Compliance With Llm Reviewing Policy:**

Affirmed.

**Final Justification:**

************* After Rebuttal *****************
I keep my positive score, and increased my confidence by one point since it looks like no other reviewer has been able to find any major issues with the originality of the results. (Something I would not be able to verify myself)

**Key Questions For Authors:**

None at this point.

**Limitations:**

Yes

**Strengths And Weaknesses:**

Strengths:
- Considers the combination of two important settings in OCO
- Shows its versatility by applying it to two other, previously studied settings, and showing that this paper's regret bounds extend and improve previous results

Weaknesses:
- None that I found. Though I am not an expert in this field, and am not able to judge whether the exact assumptions and quantities studied in this are considered appropriate. It looks like a good paper to me.

Questions:
- In lines 300 and 318, you claim near-optimality of your regret bounds. I must have missed it but could you elaborate on the near-optimality and where you derive that from?

Minor Comments:
- References in lines 128-130 were not embedded well into the sentence flow

---

> ### Author Rebuttal · Authors · 2026-03-30
>
> Thank you for putting in the effort to understand our paper, and for the
> positive review. Below we address the main question.
>
> **Q1:** In lines 300 and 318, you claim near-optimality of your
> regret bounds. I must have missed it, but could you elaborate on the
> near-optimality and where you derive that from?
>
> **A1:** Our claims of near optimality are derived from existing
> lower bounds from the easier bounded domain settings. Regarding line
> 300, for OCO with movement cost over a bounded domain, Theorem 5
> in [1] establishes the minimax optimality of the bound
> $\Theta\left(\sqrt{\lambda T(1+P_T)}\right),$ showing that it is optimal
> in its dependence on the fixed switching-cost coefficient $\lambda$ over
> rounds, the time horizon $T$, and the path length $P_T$. Regarding line
> 318, there is a lower bound of $\Omega(B\sqrt{T})$ for the static regret
> of OCO with memory; see Theorem 3.4 in [2], and the bound could be
> extended to $\Omega(B\sqrt{(1+P_T) T})$ dynamic regret using standard
> arguments (e.g., [4] Theorem 2). In addition, for OCO with fixed delay
> over a bounded domain, Theorem 3.8 in [3] provides a worst-case lower bound of
> $\Omega\left(\sqrt{d_{\max}T(1+P_T)}\right)$ for the dynamic regret,
> where $d_{\max}$ is the maximum delay. Thus the rates we obtain match
> the optimal dependencies in each of these applications in the worst
> case, but can be much smaller in practice due to being adaptive to the
> instance-specific quantities.
>
> **References:**
>
> [1] Zhao, P., Yan, Y.-H., Wang, Y.-X., and Zhou, Z.-H. Non-stationary
> online learning with memory and non- stochastic control. JMLR, 2023.
>
> [2] Raunak Kumar, Sarah Dean, and Robert Kleinberg. Online Convex
> Optimization with Unbounded Memory. NeurIPS, 2023.
>
> [3] Wan, Y., Yao, C., Song, M., and Zhang, L. Non-stationary online
> convex optimization with arbitrary delays. ICML, 2024.
>
> [4] Zhang, Lijun, Shiyin Lu, and Zhi-Hua Zhou. Adaptive Online
> Learning in Dynamic Environments. NeurIPS, 2018.

---

> > ### Author Rebuttal · Reviewer_1Aes · 2026-04-04
> >
> > Thank you. It would be good to include this info somewhere in the paper for completeness. Good luck!

---

### Official Review · Reviewer_YCoU · 2026-03-13

**Soundness:** 4
**Presentation:** 4
**Significance:** 3
**Originality:** 3
**Overall Recommendation:** 4
**Confidence:** 4

**Summary:**

The paper considers the online convex optimization problem with movement costs where the coefficient of the movement cost can vary with time and design parameter free algorithms whose dynamic regret is adaptive to (i) comparator sequence, (ii) realized gradients, and (iii) movement cost coefficients.

The main contribution of the paper is an algorithm that achieves a dynamic regret of $\tilde O (\sqrt{(M+P_T)\sum_t ((||g_t||^2 + \lambda_t||g_t||) ||u_t|| ) })$. The paper gives two applications of this framework - first to the problem of online learning with delayed feedback. Here, the paper shows a reduction from the delayed feedback setting to the setting with movement costs (with time varying coefficients) and gives improved parameter free dynamic regret guarantees.  The second application is to online learning with time varying memory.

**Compliance With Llm Reviewing Policy:**

Affirmed.

**Key Questions For Authors:**

Q. Is the OCO with time varying movement costs problem also studied in the constrained case (say the domain is the unit ball)? Add a discussion to the paper for completeness.

Q. Does one get improved rates for strongly convex functions?

**Limitations:**

yes

**Strengths And Weaknesses:**

The paper is well written and well structured. For the main result on time varying movement costs, I appreciate that the paper first presents the simple(r) base algorithm and then gradually introduces the meta-algorithm (exponential grid search) and finally the adaptive minibatched algorithm. The presentation is very clear and the paper clearly motivates each level of complexity added.

I found the paper interesting and feel that it makes an interesting contribution. The main result for time varying movement costs is not super surprising and the techniques utilized rely heavily on prior work (esp. Jacobsen & Cutkosky). Nevertheless I found the overall results neat and the reductions from delayed feedback and memory settings to be important.

---

> ### Author Rebuttal · Authors · 2026-03-30
>
> Thank you for taking the time to understand our paper, and acknowledging
> the importance of the reductions to delayed feedback and time-varying
> memory.
>
> We would like to gently push back on the claim that the "The main result
> for time varying movement costs is not super surprising and the
> techniques utilized rely heavily on prior work". We are able to borrow
> some analysis from Jacobsen & Cutkosky (2022) because both analyses
> apply mirror descent with a log-linear regularizer. The novelty of our
> analysis lies in the observation that the movement costs can be absorbed
> into the usual stability analysis and controlled alongside
> $\sum_t\langle g_t, w_t-w_{t+1}\rangle$ using the negative divergence term.
> Surprisingly, this seems to be a new insight that has not previously
> been exploited in previous works.
>
> In prior works studying the bounded domain setting, the standard
> approach to controlling the movement costs is to observe that if we use
> a gradient-descent-type algorithm, we are able to upper bound
> $\lambda \sum_t \\|w_t-w_{t-1}\\|$ by $\eta\lambda\sum_t\\|g_t\\|$ where
> $\eta$ is the learning rate, but this does not work for
> comparator-adaptive algorithms in unbounded domains, where
> $\\|w_t-w_{t-1}\\|$ can be of order $2^t$ in general (see, e.g., Lemma 8 in [6]).
> In unbounded domains, the only known approach prior to our own
> work is [5], which uses a complicated dual-space scaling argument that
> does not generalize to dynamic regret. Our approach, in contrast, is
> both simple and effective, and can be easily applied to dynamic regret.
> This is actually arguably quite surprising given the simplicity of our
> approach and the limitations of the existing techniques. We believe this
> makes our approach a solid contribution that will be of interest to the
> community.
>
> We further address specific questions in what follows.
>
> ### Answers to Questions
>
> **Q1:** Is the OCO with time varying movement costs problem also
> studied in the constrained case.
>
> **A1:** OCO with time-varying movement costs could technically be
> considered a special case of the more general smoothed OCO problem,
> wherein the movement costs are $\sum_t c_t(w_t,w_{t-1})$ for some
> semi-metric $c_t$. However, we are not aware of any works that
> explicitly consider the case $c_t(x,y)=\lambda_t \\|x-y\\|$, and moreover,
> typically these works state the problem with a time-varying $c_t$ but
> only analyze special cases with a fixed metric, such as
> $c_t(x,y)=\lambda\\|x-y\\|$ for all $t$.
>
> In the constrained setting, the closest work to ours is [4]. They
> establish that a regret dependence of $\Theta(\sqrt{G\lambda(1+P_T)T})$
> is minimax optimal when $\lambda_t = \lambda$ and $\\|g_t\\| \le G$ for
> all $t \in [T]$ (see Theorem 5 in [4]), but they do not extend their
> analysis to time-varying coefficients. However, our bound captures a
> more adaptive form of this dependency, since in the worst case we have
> $\sqrt{(1+P_T)\sum_t\lambda_t\\|g_t\\|}\le \sqrt{(1+P_T)G\lambda T}$ when
> $\lambda_t=\lambda$ for all $t\in[T]$.
>
> **Q2:** Does one get improved rates for strongly convex functions?
>
> **A2:** We are not aware of any works achieving improved dynamic
> regret guarantees under strong convexity in the general unconstrained
> setting we consider here, and it is currently unclear whether the rates
> can improve under strong convexity; this is in fact an ongoing direction
> of follow-up work.
>
> Generally speaking, even with curvature, the dynamic regret does not
> improve from $\sqrt{(1+P_T)T}$ unless one can achieve a poly-logarithmic
> strongly-adaptive guarantee (e.g. $O(\log(b-a))$ on each time interval
> $[a,b]$ simultaneously). In these cases it is often possible to show
> that the poly-logarithmic strongly-adaptive guarantee implies dynamic
> regret of order $\tilde O(P_T^{2/3}T^{1/3})$, but typically these
> results also require additional assumptions as well, such as a bounded
> domain or restricting to specific loss functions like the squared loss
> (see, e.g, [1,2,3]).
>
> **References:**
>
> [1] Baby, Dheeraj, and Yu-Xiang Wang. \"Online forecasting of
> total-variation-bounded sequences\". NeurIPS, 2019.
>
> [2] Baby, Dheeraj, and Yu-Xiang Wang. \"Optimal dynamic regret in
> exp-concave online learning\". COLT, 2021.
>
> [3] Baby, Dheeraj, and Yu-Xiang Wang. \"Optimal Dynamic Regret in
> Proper Online Learning with Strongly Convex Losses and Beyond\",
> AISTATS, 2022
>
> [4] Zhao, P., Yan, Y.-H., Wang, Y.-X., and Zhou, Z.-H.
> \"Non-stationary online learning with memory and non-stochastic
> control\". JMLR, 2023.
>
> [5] Zhang, Z., Cutkosky, A., and Paschalidis, Y. Optimal comparator
> adaptive online learning with switching cost. NeurIPS, 2022.
>
> [6] Zhang, Jiujia, and Ashok Cutkosky. \"Parameter-free regret in high
> probability with heavy tails\". NeurIPS, 2022.

---

> > ### Author Rebuttal · Reviewer_YCoU · 2026-04-04
> >
> > Thanks for the response. I maintain my (already positive) score.

---

### Official Review · Reviewer_4tqN · 2026-03-23

**Soundness:** 3
**Presentation:** 3
**Significance:** 3
**Originality:** 4
**Overall Recommendation:** 4
**Confidence:** 2

**Summary:**

The authors in this paper study unconstrained online convex optimization (OCO) with time-varying movement costs and analyze comparator-adaptive dynamic regret in this setting. The main technical result is a parameter-free algorithm achieving a regret bound with adaptivity to comparator norm/path length, realized gradients, and the time-varying movement coefficients. A key refinement is an adaptive batching reduction that improves the dependence on movement costs from quadratic to a first-order interaction term. The paper then applies this framework to two settings: delayed feedback and OCO with time-varying memory, obtaining unconstrained comparator-adaptive guarantees in both cases.

Prior work treated unconstrained movement-cost OCO, dynamic regret with movement costs, and parameter-free dynamic regret in related settings mostly separately; this submission targets their intersection and extends it to time-varying movement penalties. The delayed-feedback reduction converts delay into a movement-cost term that varies with the number of outstanding gradients.

**Compliance With Llm Reviewing Policy:**

Affirmed.

**Ethical Review Concerns:**

None.

**Final Justification:**

************* Post Rebuttal ***************** I keep my positive score.

**Key Questions For Authors:**

(1)  As mentioned in the weaknesses segment about the optimality aspect: could the authors clarify which parts of these claims are formally justified for the paper setting, and which parts are inherited from known special cases?

(2) The authors explain how to remove some prior-knowledge assumptions via a doubling trick. Could they give a concise end-to-end summary of what quantities, if any, must be known in advance in the final delayed-feedback and time-varying-memory algorithms, and where doubling/restarts are needed?

(3) In Theorem 5.7, the final guarantee depends on $\sum_t b_t^2$. Is this dependence believed to be unavoidable in the time-varying-memory setting, or is there reason to expect a sharper dependence under additional structure?

**Limitations:**

Direct negative societal impacts appear limited for this theoretical work, however, the paper would benefit from a brief explicit discussion of its technical limitations, especially the scope of the “optimal/near-optimal” claims and the assumptions needed in the final adaptive algorithms.

**Strengths And Weaknesses:**

Strengths:
* The paper addresses an interesting setting at the intersection of unconstrained OCO, dynamic regret, and time-varying movement costs, which appears not to have been studied directly before.
* The technical refinement from quadratic dependence on movement costs to a first-order interaction term is meaningful and seems especially important for the delayed-feedback application.
* The reductions to delayed feedback and time-varying memory are conceptually clean and broaden the relevance of the main result.
* The paper is well written. It is well motivated, and the discussion of prior work is comprehensive.

Weaknesses:
* The authors recover known special cases when $\lambda_t = 0$ and explain why full second-order adaptivity is impossible with movement costs, however, the paper does not provide a formal lower bound for the new dynamic setting with time-varying movement costs. As a result, it remains somewhat unclear how the optimal the presented upper bounds generally.

* This might be a minor suggestion: The technical story is interesting, but the presentation moves quickly through several layers: the base movement-cost method, the parameter-free meta-algorithm, the refinement leading to the first-order bound, and then the two application-specific wrappers. A brief roadmap paragraph explaining how these pieces fit together would make the paper easier to follow.

* The paper would be stronger with a short discussion or illustrative example explaining when the improved first-order dependence on movement costs should materially change behavior relative to previous bounds and/or atleast some synthetic experiments.

---

> ### Author Rebuttal · Authors · 2026-03-30
>
> Thank you for the detailed comments. Below, we first address a remark
> made in the weaknesses section, and then answer each of the questions.
>
> **Explanation of First vs. Second-order bounds (weakness 3):**
> Note that the first-order dependence can *always* be upper bounded in
> terms of the second-order one by AM-GM inequality:
> $\sum_t \\|g_t\\|^2+\\|g_t\\|\lambda_t\le O(\sum_t\\|g_t\\|^2+\lambda_t^2)$.
> However, the converse is not true, and the gap between these two can be
> very significant. In particular, when the gradients $\\|g_t\\|$ are close
> to $0$ while the movement costs $\lambda_t$ are large, the first-order
> bound is much better than the second-order one.
>
> A convenient illustrative example where the second-order bound is
> significantly worse than the first-order can be seen in our results for
> delayed feedback. In particular, in our reduction for delayed feedback,
> $\lambda_t=G|m_t|$ where $|m_t|$ is the number of missing observations
> at the beginning of $t$. In this case, the first-order dependencies
> scale as
> $\sqrt{\sum_t \\|g_t\\|\lambda_t}\le G \sqrt{\sum_t |m_t|}\le G\sqrt{d_{tot}}$.
> This yields an overall rate of
> $\sqrt{\sum_t \\|g_t\\|^2 + \lambda_t\\|g_t\\|}\le G\sqrt{T+d_{tot}}$, which
> is known to be minimax optimal with respect to $T$ and $d_{tot}$.
> Conversely, the second-order dependence would scale with
> $G\sqrt{\sum_t |m_t|^2}$, a bound that can easily become much worse than
> $G\sqrt{T}$ when $d_{tot}=\Theta(T)$. For instance, if all $g_t$'s where
> $t\leq \sqrt{T}$ are delayed for $\sqrt{T}$ rounds, the second-order
> term $G\sqrt{\sum_t|m_t|^2}$ becomes $O(GT^{3/4})$, while the
> first-order bound remains $O(G\sqrt{T})$. We can add a remark discussing
> this illustrative example for the next revision of the paper.
>
> ### **Answers to Questions**
>
> **Q1:** lower bounds
>
> **A1:** While we do not have a formal lower bound for our setting,
> our results match several known minimax optimal bounds from the
> literature. In particular, it is known that there is an unavoidable
> penalty of $\Omega(\sqrt{G\lambda(1+P_T)  T})$ for OCO with fixed
> movement cost $\lambda$ (see [1] Theorem 5), which matches our result
> in the worst case (i.e., where $\lambda_t\le\lambda$ for all $t\in[T]$).
> For OCO without movement costs, it is also known that there is an
> unavoidable penalty of $\Omega(G\sqrt{(1+P_T)T})$ in the worse case for
> dynamic regret in OCO. Taken together, our results capture both of these
> penalties in the worst case, but can be significantly smaller in
> practice due to being adaptive to the individual $\\|g_t\\|_2$ and
> $\lambda_t$.
>
> **Q2:** Explanation of the doubling trick
>
> **A2:** As discussed in Remark 4.2, our Algorithm 3 requires setting
> $L \geq G + \lambda_{\max}$, where $\lambda_{\max}$ is the maximum movement cost.
>
> In the delayed-feedback setting, the effective movement cost is defined
> by the number of missing gradients, meaning that
> $\lambda_{\max} = G \max_{t \in [T]} |m_t|$. This quantity is not
> necessarily known at the start of the game, but we can employ a standard
> doubling trick to estimate it on-the-fly: starting from an initial guess
> of $\hat\lambda_{\max}=G$, at the beginning of each round $t$ the
> learner can check if $G|m_t|$ exceeds our current working estimate for
> $\hat\lambda_{\max}$; if so, we can simply double the estimate (and
> correspondingly update $L$) and restart the base algorithm. A standard
> argument shows that this preserves the bound that would be obtained with
> prior knowledge of $\lambda_{\max}$ up to a constant multiplicative
> factor.
>
> Note that in the time-varying-memory setting, we have
> $\lambda_{\max} = G \max_{t \in [T]} \xi_t$, where $\xi_t$ is defined in
> Lemma 5.6. Because the sequence of memory lengths $(b_t)\_{t=1}^T$ (and
> consequently $\xi_t$) is assumed to be known to the learner in advance,
> $\lambda_{\max}$ can be computed ahead of time. Therefore, no doubling
> trick or restarting procedure is required for this application.
>
> **Q3:** Dependency on $\sum_t b_t^2$
>
> **A:** In OCO with memory, when the memory has a fixed length $B$
> (i.e., $b_t=B$ for all $t\in[T]$), the $\Theta(B\sqrt{T})$ dependence is
> known to be minimax optimal for static regret (see, e.g., Theorem 3.4 in
> [1]). Our result thus matches this worst-case dependence since
> $\sqrt{\sum_{t}b_t^2}=B\sqrt{T}$. Furthermore, this dependency can be
> inferred for dynamic regret as well: a lower bound of
> $\Omega(B\sqrt{(1+P_T)T})$ can be obtained via standard epoch-based
> arguments that lift static regret lower bounds to dynamic ones (see
> [3], Theorem 2).
>
> We will incorporate the suggestions regarding clarity and pacing in the
> next revision.
>
> **References:**
>
> [1] Zhao, P., Yan, Y.-H., Wang, Y.-X., and Zhou, Z.-H. Non-stationary
> online learning with memory and non- stochastic control. JMLR, 2023.
>
> [2] Raunak Kumar, Sarah Dean, and Robert Kleinberg. Online Convex
> Optimization with Unbounded Memory. NeurIPS, 2023.
>
> [3] Zhang, Lijun, Shiyin Lu, and Zhi-Hua Zhou. Adaptive Online
> Learning in Dynamic Environments. NeurIPS, 2018.

---

> > ### Author Rebuttal · Reviewer_4tqN · 2026-04-05
> >
> > Thank you for your response. I hope that the authors can incorporate the remarks/comments in the draft as promised. I will keep my positive score.

---

### Decision · Program_Chairs · 2026-04-30

**Decision:**

Accept (regular)

**Comment:**

This paper proposes parameter-free algorithms for online convex optimization with time-varying movement costs, with applications to delayed feedback and memory settings. All four reviewers recommended acceptance and all concerns were fully resolved in the rebuttal.

The main concerns were: the lack of formal lower bounds for the new setting; whether the first-order bound always improves over the second-order one; and whether the technical contributions go beyond prior work. The authors addressed all of these convincingly.